# A deep learning-based computational pipeline predicts developmental outcome in retinal organoids

Cassian Afting[1,2,3], Norin Bhatti[1], Christina Schlagheck[1,2,3],
Encarnación Sánchez Salvador[1,2], Laura Herrera-Astorga[1,2,4], Rashi Agarwal[1,2],
Risa Suzuki[1,2], Nicolaj Hackert[5,6], Hanns-Martin Lorenz[5], Lucie Zilova[1],
Joachim Wittbrodt [1]*, Tarik Exner[2,5,6]*

**1** Centre for Organismal Studies Heidelberg (COS), Heidelberg University, Heidelberg, Germany,
**2** Heidelberg International Biosciences Graduate School HBIGS, Heidelberg, Germany, **3** HeiKa
Graduate School on "Functional Materials", Heidelberg, Germany, **4** Programme for the Development of
Basic Sciences (PEDECIBA), University of the Republic, Ministry of Education and Culture, Montevideo,
Uruguay, **5** Division of Rheumatology, Department of Medicine V, Heidelberg University Hospital,
Heidelberg, Germany, **6** Institute for Immunology, Heidelberg University Hospital, Heidelberg, Germany

* jochen.wittbrodt@cos.uni-heidelberg.de (JW), Tarik.Exner@med.uni-heidelberg.de (TE)

Genome Engineering, KOREA, REPUBLIC OF

**Peer Review History:** PLOS recognizes the
benefits of transparency in the peer review
process; therefore, we enable the publication
of all of the content of peer review and
author responses alongside final, published
articles. The editorial history of this article is
available here: https://doi.org/10.1371/journal.
pbio.3003597

## Abstract

Retinal organoids have become important models for studying development and
disease, yet stochastic heterogeneity in the formation of cell types, tissues, and
phenotypes remains a major challenge. This limits our ability to precisely experimen-
tally address the early developmental trajectories towards these outcomes. Here,
we utilize deep learning to predict the differentiation path and resulting tissues in
retinal organoids well before they become visually discernible. Our approach effec-
tively bypasses the challenge of organoid-related heterogeneity in tissue formation.
For this, we acquired a high-resolution time-lapse imaging dataset comprising about
1,000 organoids and over 100,000 images enabling precise temporal tracking of
organoid development. By combining expert annotations with advanced image
analysis of organoid morphology, we characterized the heterogeneity of the retinal
pigmented epithelium (RPE) and lens tissues, as well as global organoid morpholo-
gies over time. Using this training set, our deep learning approach accurately predicts
the emergence and size of RPE and lens tissue formation as well as similarities in
overall organoid morphology on an organoid-by-organoid basis at early developmen-
tal stages, refining our understanding of when early lineage decisions are made. This
approach advances knowledge of tissue and phenotype decision-making in organ-
oid development and can inform the design of similar predictive platforms for other
organoid systems, paving the way for more standardized and reproducible organoid
research. Finally, it provides a direct focus on early developmental time points for
in-depth molecular analyses, alleviated from confounding effects of heterogeneity.

**Data availability statement:** Scaled and segmented images (single slice, sum- and maximum-intensity z-projections) have been deposited at Zenodo in a custom dataset format (single-slice representations: DOI: https://doi.org/10.5281/zenodo.17202714, sum-intensity z-projections: DOI: https://doi.org/10.5281/zenodo.17205312, max-intensity z-projections: DOI: https://doi.org/10.5281/zenodo.17205321). The source data for the figures have been deposited at Zenodo as well (DOI: https://doi.org/10.5281/zenodo.18198347) and can be used in conjunction with the figure generating source code (see below). The respective links are stored in the GitHub repository (see below), but repositories can be readily accessed using the provided DOIs as described above. Usage instructions for the datasets are found in the GitHub repository. Raw image data and saliency analyses (~8 TB) are available upon reasonable request.

**Funding:** C.A. and T.E. were partially funded by the Structured Doctoral programme (Strukturiertes Doktorandenprogramm zum Erwerb des Dr. med. und Dr. rer. nat.) of Heidelberg University. L.H.A. was partially funded by postgraduate grant programmes (Beca de Apoyo a docentes para estudios de Posgrados, nivel doctorado and Forschungsstipendien - Bi-national betreute Promotionen/Cotutelle, 2023/24) of the Comisión Académica de Posgrado, Universidad de la República, and the German Academic Exchange Service (57645446). This work was made possible through the funding from the Deutsche Forschungsgemeinschaft (DFG, German Research Foundation) through Excellence Cluster "3D Matter Made to Order" (EXC-2082/1-390761711), the Forschungsgruppe FOR2509, WI 1824/9-1 and the European Research Council Synergy Grant IndiGene (number 810172) to J.W.. The authors acknowledge support by the state of Baden-Württemberg through bwHPC and the German Research Foundation (DFG) through grant INST 35/1597-1 FUGG. The authors gratefully acknowledge the data storage service SDS@hd supported by the Ministry of Science, Research and the Arts Baden-Württemberg (MWK) and the German Research Foundation (DFG) through grant INST 35/1803-1 FUGG and INST 35/1804-1 LAGG. The authors acknowledge support by the state of Baden-Württemberg

## Introduction

Retinal organoids (RO), derived from embryonic and (induced-) pluripotent stem cells of human [1], mice [2], and fish [3], have become vital models for understanding the retina in development and disease. Yet, notable challenges remain and one key challenge, that is found in RO as well as in organoid systems generally, is their intra- and inter-organoid heterogeneity, which is introduced from the very onset of their assembly as well as throughout their life cycle. This heterogeneity is believed to be caused by, among others, the stochastic nature of differentiation processes and self-organization [4,5]. Organoids subjected to identical conditions within and across experiments often exhibit markedly different outcomes, including starkly contrasting and even opposing phenotypes [6,7]. RO, specifically, may vary in tissue outcomes such as the emergence of the functionally highly relevant retinal pigmented epithelium (RPE), their cell type diversity as well as more general phenotypic outcomes like symmetry, size, and shape [8].

Some degree of variability and heterogeneity in organoids might be desirable, since it reflects the complexity of biological systems [9]. However, unintended heterogeneity essentially prohibits unconfounded studies of early development and related phenotypes in organoids. Under those conditions it is impossible to predict whether a particular phenotype will emerge, and destructive analysis would preclude subsequent verification.

There is considerable interest in the (retinal) organoid community to conduct molecular analyses at single-cell resolution on organoids of different developmental stages to unveil their specific cellular composition and characteristics of developmental trajectories [10,11]. These approaches have even been combined in a multimodal manner with immunofluorescence imaging, profiling RO heterogeneity spatio-temporally within and across samples [12]. *A priori* knowledge about whether a specific phenotype will emerge in any specific organoid within an experiment would consequently enable these molecular analyses to be purged from the confounding factor of heterogeneity.

Further, it is crucial to determine at which specific time windows a trajectory towards a specific phenotypic outcome is set during organoid development. Identifying these fate-defining windows results in a better understanding of the system and allows targeted efforts in improving culture conditions. Importantly, it will guide researchers to identify promising temporally matching targets specifically triggering the developmental routes for their tissue outcomes of interest (TOI). Ultimately, this would facilitate more consistency in studying these phenotypic outcomes, thereby enhancing the utility of organoid systems as organ and tissue models in development and disease.

Recent advances in deep learning (DL) technology allowed the successful application of DL models to accurately segment organoids from images and analyze their structural features [13–16]. Further, one study effectively trained a neural network to predict mRNA expression levels in relation to tissue differentiation in kidney organoids [17], while another one successfully predicted the spatial distribution of marker genes within RO [18], demonstrating the potential of DL to bridge imaging and

through bwVisu. The funders had no role in study design, data collection and analysis, decision to publish, or preparation of the manuscript.

**Competing interests:** The authors have declared that no competing interests exist.

**Abbreviations:** AUC, area under the curve; CML, classical machine learning; CNN, convolutional neural networks; DL, deep learning; RO, retinal organoids; ROIs, regions of interest; RPE, retinal pigmented epithelium; SEM, standard error of the mean; TOI, tissue outcomes of interest.

molecular data. A convolutional neural network (CNN) was successfully employed to recognize the presence of retinal precursor structures in early retinal mouse organoids to predict later retinal identity, further underscoring the utility of these techniques in developmental biology [19]. When it comes to supervised DL in organoid research, the acquisition of annotated high-quality datasets spanning up to hundreds of thousands of images has been recognized as one of the major limitations in this area [20]. In mouse and human RO, it typically takes weeks and months, respectively, until first signs of differentiated retinal cell types can be detected [2,10]. This creates a substantial time investment when used for proof-of-concept studies, where experimental turnovers are typically high and hard to anticipate. Therefore fast-developing, non-mammalian, yet vertebrate organoid systems, like those derived from fish, that recapitulate all key developmental aspects of mammalian organ development are particularly suited for these types of studies [3,21].

Overall, despite these advances in DL powered organoid analysis and the promises DL holds when it comes to recognition of image patterns beyond human capabilities [20], there remains a notable gap in the application of DL to predict whether and to what extent tissues and phenotypes will develop within organoids.

In this proof-of-concept study, we demonstrate how DL can predict tissue outcomes in medaka fish RO well before these outcomes visibly emerge. To achieve this, we acquired a time-lapse bright-field image data set spanning 988 organoids and 117,249 images. Using expert annotation and advanced image analysis of organoid morphology, we characterize the inter- and intra-RO heterogeneity for two TOI, namely RPE and the lens, as well as for the organoids' global morphology over time. Finally, we utilize DL to break through this heterogeneity and accurately predict both the emergence and the sizes of RPE and lens tissue formation as well as the organoids' overall morphological similarities at very early developmental time points on an organoid-by-organoid basis. This substantially refines our understanding of the timeline when these tissue outcomes are determined during RO development. Our approach shows great potential for advancing our understanding of tissue and phenotype decision-making in organoid development across types and species. Moreover, it enables access to early developmental time points for in-depth molecular analyses, free from the confounding effects of tissue outcome and phenotype specific inter-organoid heterogeneity.

## Results

### Retinal organoids display inter- and intra-experimental heterogeneity with specific tissue outcomes

In order to show that DL can predict tissue outcomes in RO well before they emerge, we first generated a comprehensive dataset consisting of longitudinal brightfield imaging of ~1,000 organoids from 11 independent experiments that would enable us to monitor tissue development at high temporal resolution. For this, single RO derived from embryonic pluripotent cells of *Oryzias latipes* were seeded in 96-well plates and high throughput time-lapse imaged every 30 min for a total duration of 72 h using an automated widefield microscopy platform (**Fig 1A** and Methods). This way, we were

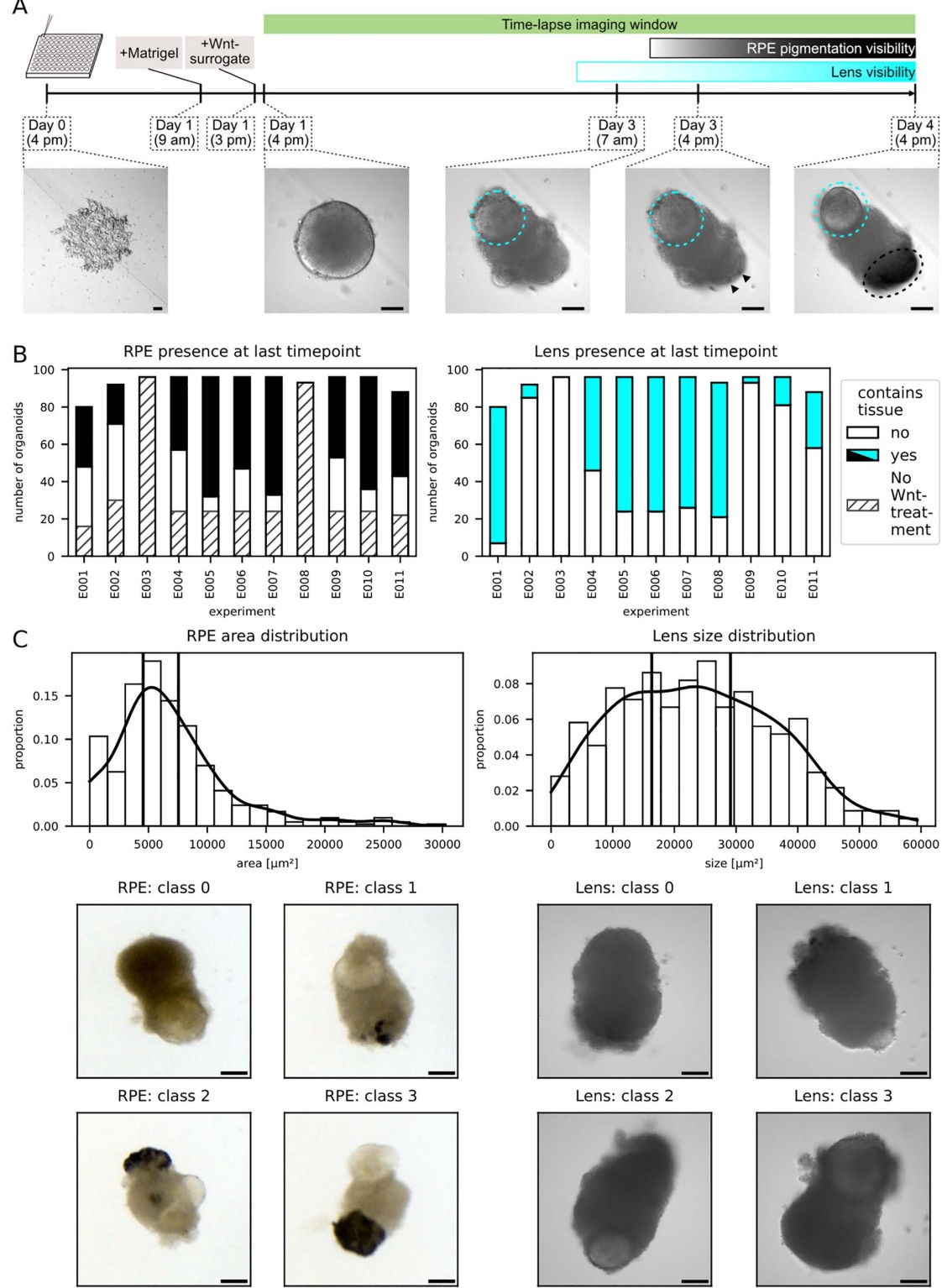

**Fig 1. Tissue-specific heterogeneity of retinal organoids across development. A** Experimental overview. Retinal organoids derived from *Oryzias latipes* embryonic pluripotent cells were treated with Wnt-surrogate 23 h after seeding and subsequently imaged by time-lapse widefield microscopy for 72 h every 30 min. The development of two tissue outcomes of interest (TOI) were followed - the lens (cyan circle) and the retinal pigmented epithelium

(RPE; marked by arrowheads and a black circle)—which begin to visibly emerge at 59 h and 68 h of organoid development, respectively. Scale bars: 100 μm. **B** Quantification of the development of the TOI. Organoids were classified for the presence or absence of RPE (left graph) or the formation of lenses (right graph) at 96 h of organoid development. Although culture conditions were kept uniform across experiments, there was high heterogeneity regarding the number of organoids developing the TOI. In the left panel (RPE emergence), the proportion of organoids without Wnt-surrogate treatment is indicated by diagonal line pattern for clarity. Lens emergence was found to be independent of Wnt-surrogate treatment and treatment conditions are thus not indicated. Raw data of the figure plots have been deposited as Extended Data 1. **C** Quantification of RPE and lens areas. Histogram of RPE (top left) and lens (top right) areas per organoid across the dataset. Four classes of TOI sizes were defined based on the quantile distributions (vertical lines; see also Methods), with organoids without development of the TOI being classified as class 0 and thus not included in the histogram. Class 1, 2, and 3 were assigned for areas below the 33rd, 66th, and 100th percentile, respectively. Example images are given for the respective TOIs (RPE: stereomicroscopy, Lens: time-lapse widefield microscopy). Scale bars: 100 μm. Raw data of the figure plots have been deposited as Extended Data 1.

able to assemble a dataset totaling 988 organoids with 117,249 images. As specific TOI, we chose RPE and the formation of lenses. Both tissues are highly physiologically and developmentally relevant [8,22,23] in the organoid context as well as easily observable in bright-field imaging. While RPE formation was triggered on-demand by the addition of the recombinant WNT surrogate-Fc Fusion Protein (Wnt-surrogate) [21] and therefore served as an example for an induced tissue formation, lenses were found to develop independently of Wnt-surrogate supplementation to the media (S1A and S1B Fig). Thus, they served as an example of a spontaneously emerging TOI in the organoids.

To delineate and cross-validate the visibility of the respective tissues over time, we first assembled an expert panel of six researchers with expertise in retinal organoid biology. The expert panel annotated a randomly selected, balanced subset of images obtained from every single organoid (see S1 Table and Methods). While the emergence of RPE became apparent at approximately 44 h into the imaging window (68 h of organoid development; Figs 1A and S1C), lenses were confidently detectable approximately 35 h after the onset of image acquisition (59 h of organoid development; Figs 1A and S1D).

Next, we set a ground truth for the presence or absence of RPE and lenses, respectively. For this, all organoids from the whole dataset were classified independently by two experts from the expert panel at the last time point imaged (96 h of organoid development). The presence of RPE was determined using stereomicroscopy due to its increased sensitivity for RPE detection compared to the time-lapse imaging data, which was confirmed by the expert panel (Figs 1A and S1C and Methods). Lens formation was evaluated for each organoid from a z-stack of the time-lapse imaging data (compare Fig 1A).

We found a marked inter- and intra-experimental heterogeneity regarding tissue emergence of RPE and lenses in the RO, despite having taken extensive care to maintain constant and highly reproducible culture conditions. Even the on-demand induction of RPE by Wnt-surrogate did not ascertain the emergence of RPE in every given organoid (Figs 1B, S1A; S2 Table). We quantified the RPE and lens areas in the stereomicroscopy and time-lapse images, respectively, as an approximation for the RPE amount and lens size that developed in each organoid. Based on the distribution of the quantified areas, we then grouped each organoid into one of 4 classes (Fig 1C). Class 0 denoted the absence of tissue emergence in a given organoid, while class 1, 2 and 3 were assigned for areas below the 33rd, 66th and 100th percentile, respectively. As it was the case for RPE and lens emergence, RPE amount and lens sizes showed a similarly large heterogeneity within and across experiments (S1E and S1F Fig). Moreover, increasing concentrations of Wnt-surrogate (1, 2 and 4 nM) applied did neither reliably ascertain the emergence nor increasing amounts of RPE to develop (S1A Fig and S2 Table).

These results showed that the tissue formation and sizes of both RPE and lenses in each organoid are heterogeneous and could not be ascertained, even when induced in a controlled fashion via an external stimulus.

## Development of a large-scale time-lapse image analysis platform

For the purpose of analyzing the time-lapse images on a large scale, we next developed a python-based analysis pipeline, starting with a DL guided image segmentation. The segmented images were processed through our analysis platform, enabling us to quantify shape descriptors and image moments, among others, over time (total number of

parameters: 165). This generated a descriptive, tabular dataset containing morphological characteristics of the organoids, termed morphometrics (Fig 2A). Distance measurements in conjunction with dimensionality reduction (Figs 2B–C, S2, and S3A–S3D) revealed that organoids were more similar to each other at early time points, but progressively diverged as development advanced (Figs 2B–C, S3E, and S3F). This suggested dynamic interindividual changes in their morphological characteristics over time (Fig 2C), which are in accordance with the literature reporting inter- and intra-RO heterogeneity on a transcriptional level over time [12]. Although global morphological changes might partially be induced by the addition of Wnt-surrogate to the culture media, as has been previously reported in organoid systems for other Wnt agonists [24], the same trends were found in RO without addition of Wnt-surrogate (S2B Fig). When we examined the morphological changes per RO over time, we could observe a considerably higher morphological change-rate in the second half of the imaging window compared to the first half, suggesting a higher organoid plasticity in later developmental stages (Fig 2D). This again was found in RO without addition of Wnt-surrogate as well (S2C Fig). Therefore, heterogeneity was not only found in a tissue-specific manner but was also reflected globally in the temporal dynamic of the RO morphology.

### Deep learning predicts RPE and lens emergence well before visibility

Following the acquisition, annotation and analysis of our time-lapse RO imaging dataset, we next focused on the image classification task aimed at predicting the emergence of RPE and lenses.

As a reference for our prediction results, the expert panel was asked to *predict* the emergence of the tissues as well as their amount at the last time point of imaging using the same data subset as described above (Fig 3A; S1 Table). Expectedly, the prediction accuracy of humans was low in the beginning and increased steadily towards later time points, consistent with the increasing visible emergence of RPE and lenses in the organoids (Fig 3B and 3C, compare S1C and S1D Fig). Notably, the F1-metric plateaued at 0.7–0.8 for the expert evaluation at later time points, which is in line with our findings regarding the visibility of the respective tissues (compare S1C and S1D Fig). Therefore, we concluded that an accurate *prediction* of tissue emergence of the TOI was infeasible for humans.

For machine-learning model training, we deliberately created two sets of test data that were not used for training. The first set, which we termed validation set, was derived from 10% of organoids that were imaged within the same experiments that were used for training. The second set, termed test set, contained organoids that were imaged in a completely independent experiment. Finally, we used a cross-validation strategy, where each of the acquired experiments was set as the test experiment once and the training was performed on 90% of organoids of the remaining experiments (Fig 3A). By using this strategy, we were able to evaluate our model's accuracy within as well as across the maximal breadth of inter-experimental variation in our data set.

In a first attempt for a machine-learning guided prediction of RPE and lens emergence, we aimed to predict the emergence of the respective tissues from tabular morphometrics data obtained from our image analysis pipeline. To facilitate this, we benchmarked several machine learning classifiers via conventional cross-validation in conjunction with hyperparameter tuning of selected classifiers to select the best performing one for the image classification task (S4 Fig and Methods). In order to capture whole organoid image information obtained from 5 z-slices spaced 50 µm around the focus plane, we calculated sum- or maximum-intensity projections. Classifier hyperparameter tuning obtained from morphometrics calculated these projections highlighted similar suitable classifiers (S5 and S6 Figs). Finally, a Random Forest classifier and Quadratic Discriminant Analysis were chosen for the prediction of RPE and lens emergence, respectively.

RPE emergence was initially predicted at F1-score accuracies of 0.65–0.75 with this model. The accuracy slightly increased between 30 and 45 hours (42%–63% of the imaging window) to a F1-score of 0.8, as time points approached the first visibility of the tissue. Notably, the accuracy was slightly outperforming the human prediction for all time points (Figs 3B, S1C, S7A, and S7B). These results indicated that the classifier trained on tabular image analysis data reliably recognized tissue presence and was somewhat able to predict RPE emergence beforehand, yet only slightly outperforming human predictions. Although initially predicting lens emergence at random, we found a slight, but distinct increase in

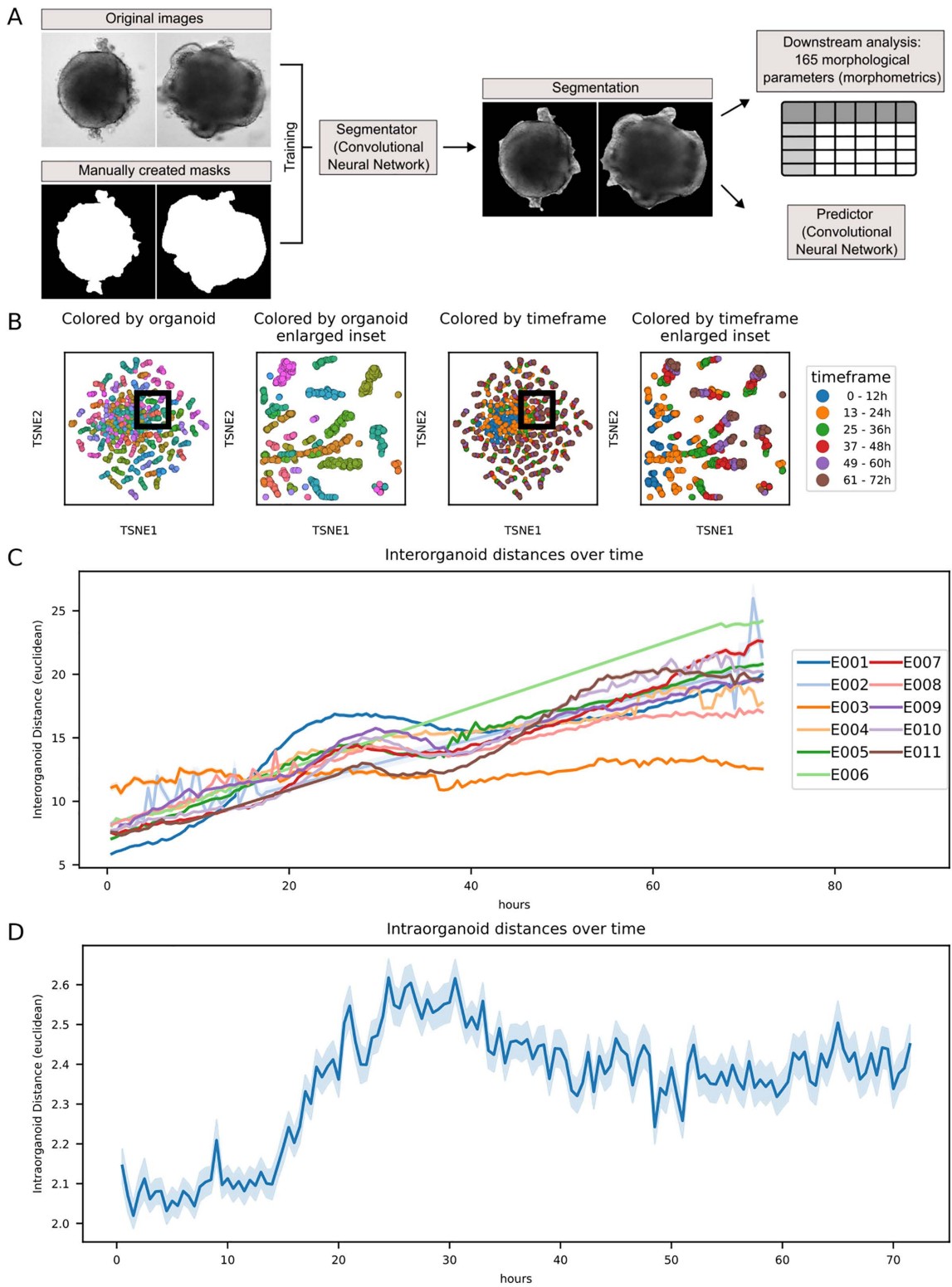

**Fig 2. Global morphological heterogeneity of retinal organoids across development. A** Image segmentation and analysis pipeline. A convolutional neural network (CNN) with the DeepLabV3 architecture was trained on 841 images and their manually annotated masks (left). The CNN was subsequently used to segment all images of the dataset that were finally subjected to a pipeline extracting a total of 165 morphological parameters

(morphometrics, right), including, among others, shape descriptors and image moments. **B** Intra-experimental global morphological organoid heterogeneity. Time-series images from one representative experiment were analyzed using the image analysis pipeline and subjected to t-SNE dimensionality reduction on the first 20 principal components. Data points were colored by individual organoids (left graph) and time frames of organoid development within the imaging window (right graph). While organoids clustered closely at earlier time points (up to 24 h), they strongly diverged at later time points, suggesting increasing inter-individual changes of their morphological characteristics over time. Raw data of the plots have been deposited as Extended Data 2. **C** Global morphological inter-organoid heterogeneity across experiments and time. Euclidean distances were calculated based on 20 principal components derived from the morphometrics data for each organoid and plotted as the mean pairwise euclidean distance between all organoids for each time point. Notably, inter-organoid distance increased over time in an experiment-specific manner, indicating an increasing morphological divergence of the individual organoids over time. Raw data of the plots have been deposited as Extended Data 3. **D** Intra-organoid morphological changes over time. Euclidean distances were calculated based on 20 principal components derived from the morphometrics data for each organoid between time point $n$ and time point $n+1$. The resulting metric reflects the amount of morphological changes during a time span of 30 min. While the relative changes are comparatively small at the beginning, the increase over time is suggesting more drastic morphological changes at later time points, consistent with the findings described in **B** and **C**. Shaded areas represent the standard error of the mean (SEM). Raw data of the plots have been deposited as Extended Data 3.

prediction accuracy to an F1-score of 0.6–0.7 with the model between 15 and 25 hours (21%–35% of the imaging window). However, classification based on the morphometrics data was found to be inferior to the human prediction at later time points, suggesting that the morphometrics data did not adequately capture features associated with lens tissue emergence (Figs 3C, S7C, and S7D). Using the sum- or maximum-projections as described above did not significantly alter the results (S8–S10 Figs).

In a second attempt, we trained an ensemble of CNNs to classify images into two categories, predicting the final presence or absence of RPE and lenses at the last time point. The CNNs demonstrated stable improvements during training as judged by the increase of the F1 metric (S11 Fig).

Strikingly, the DL model accurately predicted RPE and lens formation even at very early developmental stages (Figs 3B, 3C, and S12). For the prediction of RPE, the network achieved its first substantial accuracy with a F1-metric above 0.85 at a median of 11 h after the onset of imaging, marking a critical early threshold for accurate RPE prediction (also compare S12A and S12B Fig). The F1-metric was close to 0.9 for most of the remaining time points, providing far superior results compared to both the human prediction and the classification using tabular image analysis data (Fig 3B). These results indicated that the CNN ensemble was able to predict the emergence of RPE long before visibility of said tissue (compare Figs 1A and S1C) and even outperformed humans in organoids at time points when RPE was visible and detectable in stereomicroscopy but not detectable in the focus plane of the time-lapse images. For the prediction of lenses, the CNN ensemble was able to achieve its first confident prediction with an F1-metric above 0.85 at even earlier time points compared to the prediction of RPE (4.5 h). With a stable F1-score at that level over the remaining time points, the neural net ensemble clearly outperformed both the prediction capabilities of the experts and the tabular image analysis data-based model statistically significantly (Figs 3C, S12C, and S12D; S1 File). These results indicated that the prediction of lens emergence could be facilitated at very early developmental time points as well, even earlier than those of RPE using the same data set basis. Using sum- and maximum-intensity z-projections of the organoids as training images did not further improve the prediction accuracy for this task as well (S8, S13, and S14 Figs).

When we compared the performance on the validation and test sets as described above, we noticed that the models would generally perform more accurately on the validation organoids, which were derived from the same experiments as the training set, compared to the test organoids which were acquired in an independent experiment (Figs 3B, 3C, S7, S8, S9, S10 and S12–S14). Additionally, we observed a substantial variability in the prediction accuracy in some test experiments (S7, S8, S9, S10 and S12–S14 Figs), underscoring the inter-experimental heterogeneity of organoid development and the necessity for our cross-validation strategy (Fig 3A).

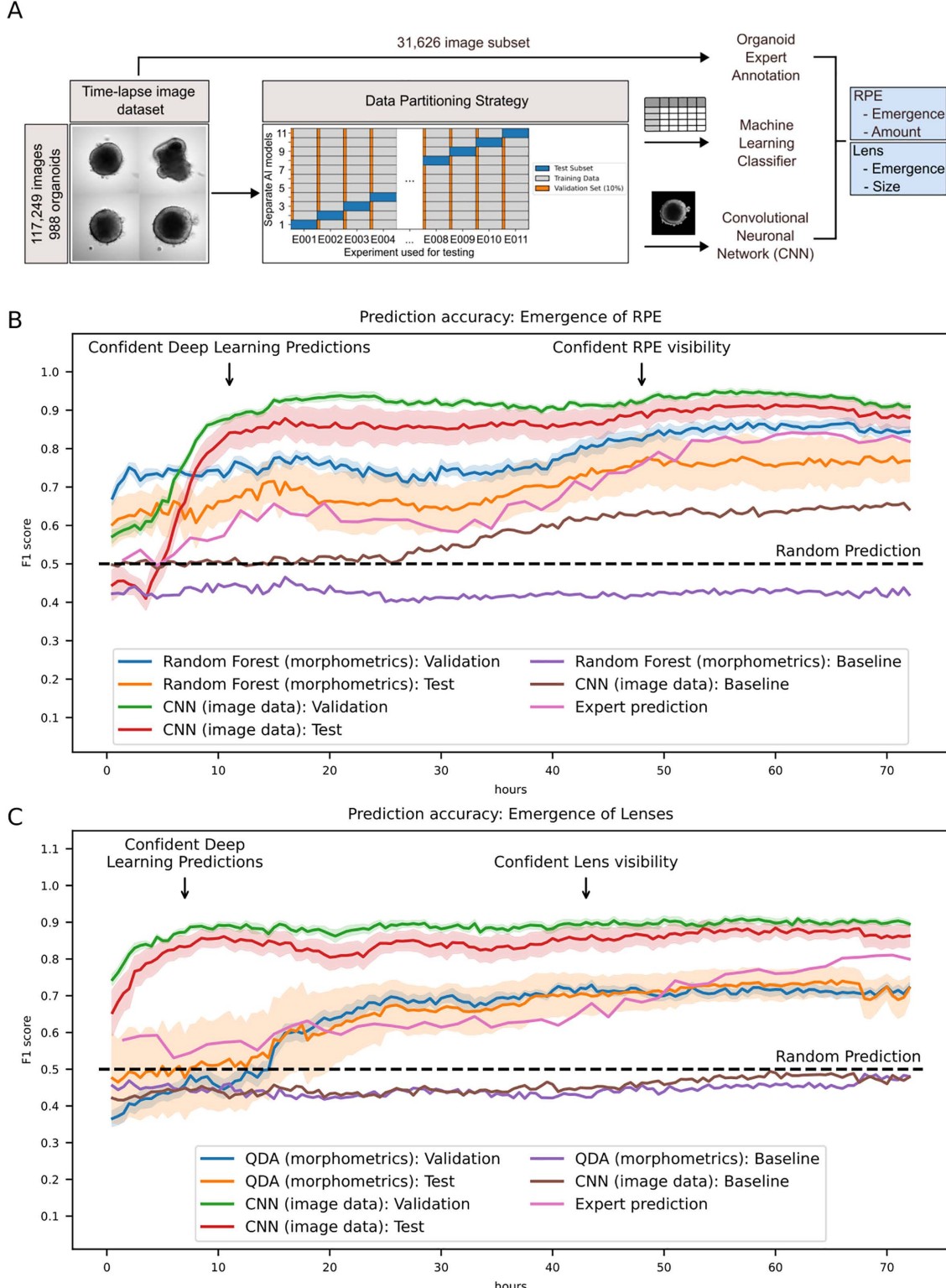

**Fig 3. Deep-learning aided prediction of RPE and lens tissue emergence in retinal organoids. A** Schematic representation of the data partitioning and machine learning model training strategy. Timelapse images were split into a training dataset and two distinct non-training datasets (data partitioning strategy): The validation set was assembled from 10% of individual organoids that were imaged during the same experiments as the training set but were not trained on. The test set was derived from organoid images acquired during a completely independent experiment that was not trained on.

This strategy was repeated in a cross validation split 11-fold. The images were either analyzed via the image analysis pipeline or subjected directly after segmentation to a CNN ensemble. For the expert annotation, a randomly selected, balanced subset of 31,626 images from across all time points in the full image dataset was designated as an evaluation set for classification and prediction (compare S1 Table and Methods). **B/C** Prediction of RPE (B) and lens (C) emergence by deep learning. The indicated classifiers were trained on the segmented image data and morphometrics tabular data, respectively, and evaluated on the validation and test set as described in A and Methods. While the classifiers trained on the tabular data could reproduce (C) and even slightly outperform (B) the human prediction accuracy, suggesting mostly tissue recognition, the CNN ensemble is able to predict RPE (B) and lens (C) emergence well before visibility at 11 h and 4.5 h, respectively (F1-score > 0.85). Lines represent the mean F1-score per time point, while shaded areas represent the standard error of the mean (SEM) over all validation-experiments and test-experiments, respectively. SEMs have been omitted for the human evaluators and the baseline (control) models for better visibility. Horizontal dotted lines highlight the hypothetical accuracy of random predictions while baselines show the F1 accuracy of models trained on the same image data with randomly shuffled labels. QDA: Quadratic Discriminant Analysis. Raw data of the plots have been deposited as Extended Data 4 and 5, respectively. Predictions were calculated using the function 'get_classifi-cation_f1_data' of the module orgAInoid.figures.figure_data_generation (compare source code).

## Tissue size prediction by deep learning

Next, we repeated our analyses attempting the prediction of the area of the RPE and lenses. As described above, we discretized the area obtained from stereomicroscopy and time-lapse image data into 4 classes based on the area distribution over all conducted experiments (Fig 1C; S2 Table).

We followed the same steps as above, including the human reference prediction, the classifier benchmark and hyper-parameter tuning on single slice images as well as sum- or maximum-intensity z-projected images (S15–S17 Figs) for the prediction from tabular image analysis data. The accuracy of the human prediction of tissue sizes increased steadily over the course of organoid development, consistent with findings obtained from the tissue emergence experiments. Human prediction reached F1-scores of 0.3–0.4 early on with a maximum F1-score of ~0.53 for RPE areas (Fig 4A) and ~0.65 for lens sizes (Fig 4B) towards the end of the imaging window after the onset of tissue visibility. This suggests that prediction of the amount of tissue is largely infeasible for humans.

We next trained a HistGradientBoostingclassifier and a Quadratic Discriminant Analysis for the prediction of RPE and lenses, respectively, and observed a similar steady increase of prediction accuracy as judged by the F1-metric compared to the human prediction. The F1-score reached a plateau which was found to be slightly above the human performance for RPE at all time points and slightly inferior for lenses at later time points only (Figs 4 and S18).

When we trained a CNN ensemble (S22 Fig), we observed a spike in prediction accuracy at very early developmental time points in line with our results obtained from the RPE emergence (compare Fig 3B) and the lens emergence (compare Fig 3C) with a final F1 metric of >0.7 for both tissues. Prediction accuracies were again stable after the initial spike, as it was the case for RPE and lens emergence predictions (Figs 4 and S23). Thus, for most RO the relative amount of RPE and the relative size of lenses could confidently be predicted around the same time points as their emergence, yet with lower overall accuracy. As it was the case for the RPE and lens emergence, we again noticed variance in the prediction accuracies between independent experiments (S18 and S23 Figs) and did not note any significant changes when using sum- or maximum-intensity z-projected images (S19–S21 and S23–S25 Figs).

Next, we sought to find relevant structural information in the images that would guide the DL model's decisions when predicting TOI at early time points before visibility. Across eight analyzed relevance backpropagation methods (compare Methods and S1 Note) and all three CNN architectures, we observed notable differences in how relevance was assigned and how consistently these assignments aligned. Highlighted organoid patterns were not consistent enough across attribution methods to support biological interpretation. Consequently, relevance backpropagation did not reveal clear organoid features that could explain the CNN's decision-making or provide early indicators of morphology linked to the TOI.

## Deep learning predicts organoid morphology

After showing the predictive capabilities of our models with two TOI, we next tested our approach on the intrinsic organoid morphology itself, independent of specific differentiation events as an example of a prediction task addressing

 

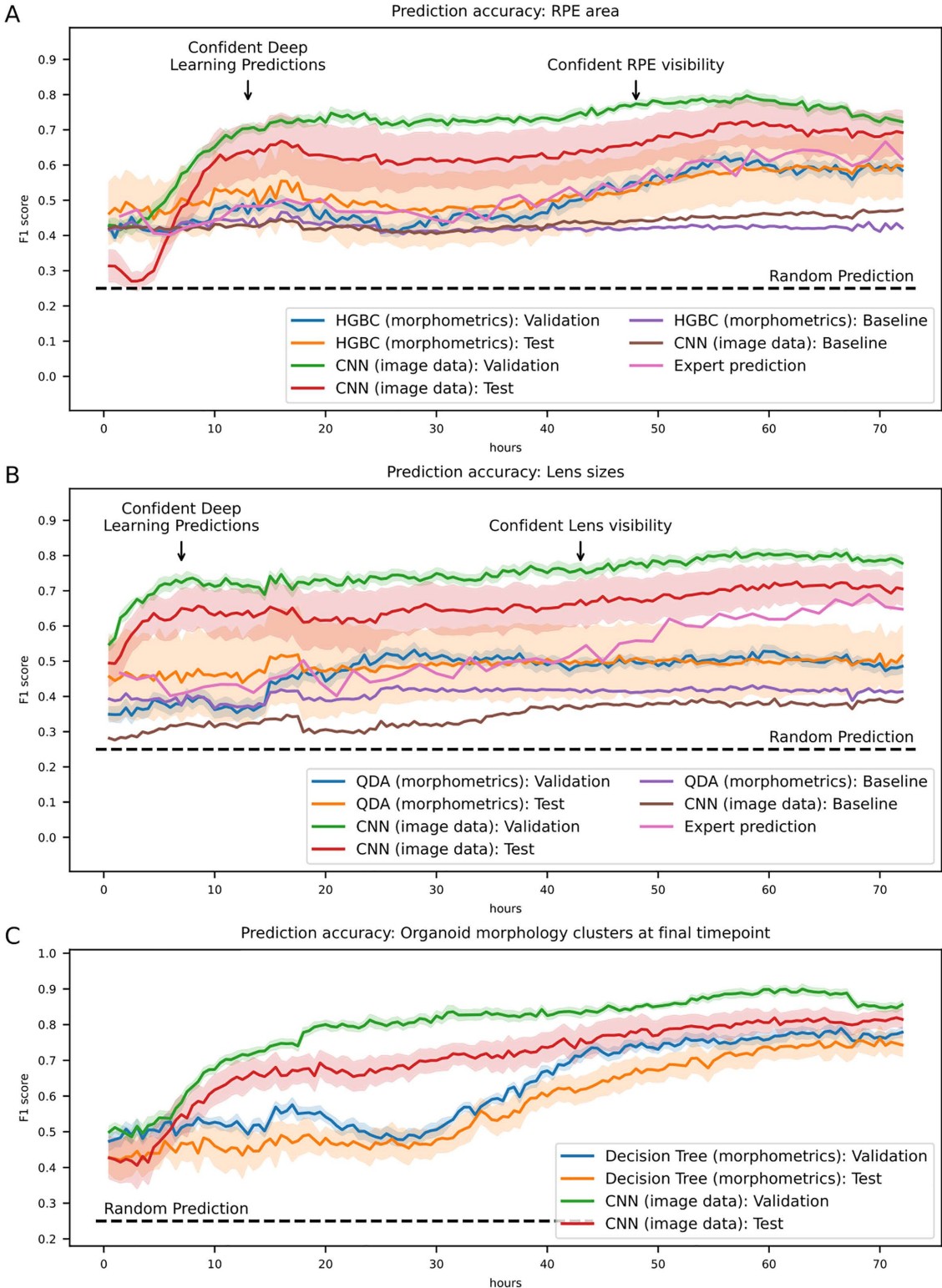

**Fig 4. Deep-learning aided prediction of RPE and lens tissue sizes as well as organoid morphology in retinal organoids. A/B** Prediction of RPE (A) and lens (B) tissue sizes by deep learning well before visibility. The indicated classifiers were trained on the segmented image data and morphometrics tabular data, respectively and evaluated on the validation and test set as described in **Fig 3A** and Methods. While the classifiers trained

on the tabular data could reproduce or slightly outperform the human prediction accuracy, the CNN ensemble is able to predict RPE (B) and lens (C) tissue sizes at substantially higher accuracies well before visibility. Lines represent the mean F1-score per time point, while shaded areas represent the standard error of the mean (SEM) over all validation-experiments and test-experiments, respectively. Horizontal dotted lines highlight the hypothetical accuracy of random predictions, while baselines show the F1 accuracy of models trained on image data with randomly shuffled labels. HGBC, HistGradientBoostingClassifier; QDA, Quadratic Discriminant Analysis. Raw data of the plots have been deposited as Extended Data 6 and 7, respectively. **C** Prediction of organoid morphology by morphometrics clustering at the final imaging time point. Organoids were clustered at the final imaging time point using morphometric features, and the resulting cluster assignments were used as categorical targets. A decision tree classifier trained on morphometric features and a CNN trained on image data were applied to predict cluster membership from earlier timepoints. Both approaches achieved substantially higher accuracy than random predictions, with CNN performance exceeding 70% F1 by ~20 h after imaging onset and the decision tree reaching similar levels later. Lines represent mean F1-score per time point, and shaded areas denote the standard error of the mean (SEM). The horizontal dashed line indicates the expected accuracy of random predictions. Raw data of the plots have been deposited as Extended Data 8. Predictions were calculated using the function 'get_classification_f1_data' of the module orgAInoid.figures.figure_data_generation (compare source code).

multi-outcome predictions of a continuous feature space. To this end, we clustered organoids based on their morphometric feature space at the final imaging time point and used these cluster assignments as categorical targets for classification. Both a decision tree classifier trained on morphometric features and a CNN trained on image data were then applied to predict the final cluster membership from earlier timepoints. Prediction accuracy, quantified by F1 score, steadily increased over time as organoids became progressively more similar to their final state (Fig 4C). Remarkably, both approaches achieved substantial accuracy much earlier than expected: CNN performance exceeded 70% F1 by approximately 20 hours after imaging onset, while the decision tree classifier reached similar levels somewhat later. This finding indicates that organoids already display early morphological signatures that are strongly associated with their eventual developmental trajectory.

## Discussion

### Model system heterogeneity as a challenge

In our study, both non-spontaneous (RPE) and spontaneous (lens) tissue emergence, as well as their final tissue sizes, were found to vary between organoids and experiments undergoing the same differentiation protocol. Despite removing technical variability to the best of our ability (compare Methods), this type of heterogeneity seemed to be a rather inherent characteristic of our model system. Researchers are in general frequently confronted with considerable heterogeneity within their organoid model systems across and within experiments. These include variation in terms of cell type diversity and patterning, reaction to external stimuli, and organoid morphology, among others [4,25]. The reasons for these heterogeneities are largely unknown, but are hypothesized to include, among others, deviations between differentiation protocols, batch-to-batch differences of cell culture material, and cell sources [25–28]. In line with this, we extend previous work [12] by demonstrating the *morphological* heterogeneity of RO within and across experiments over time using advanced image analysis tools. We comprehensively showed by distance analysis and dimensionality reduction that our model system exhibited intra- and inter-experimental variation of organoid morphology that consistently increased over time in an experiment-specific manner. While researchers in general undergo sincere efforts to minimize technical variation that would cause heterogeneity, there is yet no reproducible way to gain full control over all aspects of these complex model systems.

Due to the inherent intra- and inter-experimental heterogeneity of organoids, one can only be certain that a specific organoid possesses one TOI once it is properly and detectably established. Prior to this point, it remains uncertain whether a given organoid will develop the TOI. To describe this, we introduce the concept of the *Latent Determination Horizon* (LDH, Fig 5) as a theoretical framework derived from our observations. The LDH represents the period during which the eventual presence or absence of a TOI is not yet observable but the decision toward developing the TOI is likely being determined. This rationale stems from two main findings: (i) despite using standardized protocols, we consistently observed variable emergence of TOIs across organoids, and (ii) predictive performance of our DL models increased over time, suggesting that morphological signals carrying predictive information only become reliably available after a certain

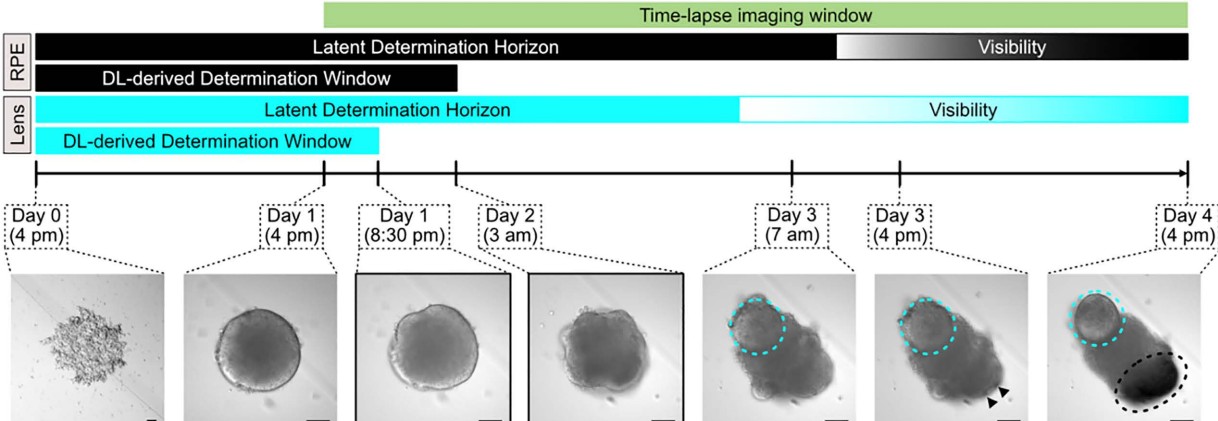

**Fig 5. Deep learning mediated predictions of tissue outcomes narrow down phenotype determination windows in organoid development.** Schematic representation of the retinal organoid developmental timeline showing the theoretical time windows (Latent Determination Horizon) during which the decisions towards RPE and lens tissue outcomes have to be made, and the narrowed-down time windows derived from deep learning (DL), where these decisions are actually being made. Lenses outlined in cyan and RPE indicated by arrowheads and a black circle. Scale bars: 100 μm.

point. Early detectable signs - such as specific transcriptional landscapes or single cells adopting the desired differentiation trajectory - may provide predictive hints but do not necessarily guarantee that the TOI will emerge. The stochastic nature of differentiation, along with unreliable spatio-temporal distribution of crucial cues (or the absence of such cues), may ultimately prevent the TOI from emerging. Moreover, some TOIs or desired phenotypes might not have any known early detectable markers.

## Comparison of machine learning approaches

To tackle the problem of organoid heterogeneity, we established a strategy that circumvents the challenge of heterogeneity within a larger group by predicting the outcome of a specific, singular organoid instead.

To accomplish that, we chose images as the fundamental, non-invasive data source, allowing studies of the respective organoids after analysis. This is in striking contrast to techniques like RNA sequencing that would require disintegration of the individual organoid and therefore prohibits studies at later time points of the same organoid. We built a high-temporal-resolution dataset of time-lapse images spanning about 1,000 organoids over the course of 72 h, which covered induction and development windows of RPE and lenses.

In order to predict tissue outcomes in a specific organoid, we chose two different approaches: on the one hand, a "classical machine learning" (CML) approach based on tabular data obtained from our analysis pipeline and on the other hand, a DL approach using the segmented images directly. When we compared the tissue outcome prediction accuracy of CML with that of the DL approach, DL consistently outperformed CML across all prediction tasks. Notably, CML did not perform significantly better than predictions made by human experts most of the time, indicating tissue recognition rather than prediction. This suggested that, for our prediction tasks, the relevant structural information within the images could not be effectively captured by existing bioimage analysis parameters or recognized by human experts through pattern recognition. In contrast, the predictive information within the time-lapse images is likely so complex and non-intuitive that only DL based on the segmented images was able to extract it successfully. Supporting this, our efforts to delineate the relevant structures for the DL classification at earlier time points were at best challenging, as we were not able to identify comprehensive organoid structures in the pixels the classifiers deemed relevant. This has been previously observed for relevance backpropagations in other contexts [29,30]. Future work should therefore move beyond pixel-level saliency maps and test

generative approaches that synthesize class-maximizing examples, and (where available) relate internal network features to molecular readouts to better connect predictions to biology. Taken together, we were able to reliably predict the emergence of RPE and lenses in individual organoids well before their actual visibility using DL.

### Deriving decision-making windows from prediction of tissue outcomes

By utilizing DL to predict tissue outcomes in organoids well before they visibly emerge, we could infer the time points in organoid development when the tissue outcome is already determined and identify those individual organoids that will adopt the TOI. This approach allowed us to narrow down the Latent Determination Horizons to *deep learning-derived determination windows* (**Fig 5**)—specific periods during which the decision toward the TOI in a given organoid is actually made. Consequently, efforts to decrease the variability of the TOI - such as modifying cell culture conditions - should be focused on these critical time frames of organoid development. However, it remains unclear whether the earliest time point at which DL can accurately predict the TOI corresponds to the actual biological decision-making moment or if it reflects a technical limitation of the model due to insufficient or inadequate training data. Therefore, the period of organoid development preceding this point should also be considered as temporal window during which decision-making occurs.

Using this methodology, we were able to substantially narrow down the decision-making window for both the emergence and ultimately the tissue sizes of RPE and lenses in RO. Interestingly, for both TOIs, these decisions coincide temporally to a large degree.

### Molecular analyses unconfounded by tissue-specific heterogeneity

One potential application of our analysis is the invasive study of tissue emergence in organoids. Consider a histological or transcriptomic study of organoids that will contain a specific tissue compared to organoids without it. As tissues have not yet emerged, grouping of the organoids can only be done at random for prospectively forming tissues. In conjunction with the invasive nature of the analysis, there is potentially no way of retrospectively identifying organoids, which would have developed the TOI and thus essentially prohibit such analyses. In line with our findings, even assigning the groups based on the on-demand induction of TOI by external stimuli would result in potentially highly inhomogeneous datasets with a strong confounding uncertainty of tissue outcome. Applying our model to predict TOI in organoids at time points where the models have shown confident prediction accuracies for the TOI will therefore substantially increase the homogeneity of the respective groups.

We deliberately selected an experimental setup that allows to make assumptions about the applicability of our models across experiments. This is especially important considering the large inter-experimental heterogeneities that were found in our dataset which we expect to extrapolate to other organoid systems. We observed a strong variability of model performance in independent experiments that would partially differ notably from the accuracy observed from organoids that were imaged during the same experiments that were used for training (i.e., validation set). Despite this variability, we are still confident that our technique can lead to a much higher purity of organoid groups compared to random group assignment. Importantly, such predictive models could also enable dynamic interventions. For example, culture conditions might be altered mid-course for organoids predicted not to form RPE, to test whether their developmental trajectory can be redirected. While speculative, this possibility highlights how predictive modeling may shift organoid research from observational strategies toward more adaptive and experimentally responsive designs.

### Applicability to other model systems and tissue outcomes

We strongly believe that our proof-of-concept study will pave the way for similar analyses across other model systems, tissue outcomes and more. In fact, our study showed that even prediction of an abstract morphological cluster is very well feasible. In order to be applicable to other scenarios, the ground-truth annotation is of high importance. Even though the annotation of dark-pigment containing RPE and relatively large lenses might seem trivial, the presence of RPE and lenses

were disagreed upon by the two independently annotating experts defining the ground truth in a considerable fraction of organoids (3.9% for RPE, 2.9% for lenses). This is an example for potential challenges that may ultimately require extensive work beforehand to ensure a highly accurate ground truth annotation. Future studies can reduce manual effort by using the CNN to pre-label high-confidence cases and by focusing expert review on uncertain or rare events. Ground truth can be defined from the final frame and a small early window where predictions are reliable, with programmatic quality checks (blur/contrast) to exclude poor frames. A second reviewer would only be needed for flagged edge cases. This model-assisted workflow is supported by our single-organoid setup and the quantitative feature set reported here. We also note that in our experimental setup a single organoid was cultured per well, which avoided the need for instance segmentation or tracking. This design choice simplified the analysis but may not always be feasible in other organoid systems, such as cancer organoids, where multiple structures per well are common and would require additional computational steps. However, our study demonstrates that the effort is worth it, even if and in particular when the development of the organoids takes extended periods of time.

We identified generalizability across laboratories as a potential limitation. Despite standardization, organoids show intra- and inter-experiment heterogeneity as is typical for these systems [4–7]. Therefore, models trained here may not reach the same accuracy on external datasets, especially with different differentiation protocols. Even so, the approach is easy to adopt, and fine-tuning our models with a small set of local images should adapt them well. Our proof-of-concept used *Oryzias latipes* derived organoids, which are reproducible and develop quickly, but species differences may affect the results. The method is not tied to *Oryzias latipes* and should extend to mammalian or human iPSC/ESC organoids if annotated datasets are available. Validation across independent organoid systems will be needed to confirm robustness and broader use. In future studies, we aim to test cross-system transfer by fine-tuning a model trained on our data with a small set of images from other labs and species (for example mouse or human organoids), to quantify how much local data is needed to reach reliable performance.

In this work, we used 1,000 organoids in total, to achieve the reported prediction accuracies. Yet, we suspect that as little as ~500 organoids are sufficient to reliably recapitulate our findings (compare S1 Note). Therefore, our approach is readily applicable to any organoid model systems of choice given the reported success rate with a limited dataset. However, even though the indicated number of individual organoids may seem small in comparison to conventional datasets in DL, we note that the required number of unique organoids may exceed feasibility for some extreme models. Beyond our current DL implementation, potential enhancements of our strategy could include the combination of tabular and image data, creating a broader dataset. Furthermore, a beneficial expansion of a time-lapse brightfield dataset might be the additional acquisition of epifluorescence images for every organoid. Apart from the additional information that transgenic reporter lines might provide, organoid autofluorescence and its spatio-temporal distribution might unlock image information beyond those obtained from bright-field imaging.

In summary, we have demonstrated that tissue and morphological trajectories in RO are reliably predicted well before the tissues and the final organoid morphology visibly emerge. We achieve this by applying a DL approach to time-lapse bright-field image datasets, thus delivering prediction results instantaneously. This predictive capability provides vital insights into the timelines of decision-making during organoid development. Our DL and data curation framework can inform the design of similar frameworks for organoids across different types and species. This approach not only unlocks these developmental insights but also crucially grants access to early developmental time points for complementary in-depth molecular analyses, which were so far confounded by heterogeneity related to the TOI.

## Methods

### Ethics statement

Medaka fish (*Oryzias latipes*) stocks were maintained according to the local animal welfare standards (Tierschutzgesetz §11, Abs. 1, Nr. 1, husbandry permit AZ35-9185.64/BH, line generation permit number 35–9185.81/G-145/15 Wittbrodt).

## Fish husbandry and maintenance

Medaka fish (*Oryzias latipes*) were kept as closed stocks in constantly recirculating systems at 28 °C with a 14 h light/10 h dark cycle. For this study, the following medaka strain was used: Cab strain [31].

## Generation of retinal organoids from medaka embryonic pluripotent cells

RO were generated from medaka embryonic pluripotent cells as previously described [3] with some changes to the differentiation protocol. Briefly, blastula-stage (6 hours post fertilization; [32]) medaka embryos were taken as a source for primary embryonic pluripotent cells, which were resuspended in modified differentiation media (DMEM/F12 ((Dulbecco's Modified Eagle Medium/Nutrient Mixture F-12), Gibco Cat#: 21041025), 5% KSR (Gibco Cat#: 10828028), 0.1 mM non-essential amino acids, 0.1 mM sodium pyruvate, 0.1 mM β-mercaptoethanol, 20 mM HEPES pH = 7.4, 100 U/ml penicillin-streptomycin) following isolation. Cells were seeded in densities of approximately 1,500 cells per organoid (approx. Fifteen cells/μl, 100 μl total) per well in low-binding, U-bottom-shaped 96-well plates (Nunclon Sphera U-Shaped Bottom Microplate, Thermo Fisher Scientific Cat#: 174925) and incubated at 26 °C without $CO_2$ control. On day 1 at 9 am, organoids were washed with differentiation media, transferred into new low-binding, U-bottom-shaped 96-well plates, and Matrigel (Corning, Cat#: 356230) was added to the media to a final concentration of 2% and a total media volume of 120 μl. Organoids were incubated for 6 h at 26 °C under $CO_2$ control. At 3 pm (day 1) WNT surrogate-Fc Fusion Protein (Wnt-surrogate; ImmunoPrecise Antibodies; Cat#: N001, Lot: 7568) was added directly into the wells to final concentrations of 1, 2, and 4 nM. Wnt-surrogate was stored in 250 nM aliquots in Wnt-surrogate dilution buffer at −20 °C and pre-diluted to 100 nM in differentiation media prior to addition to the wells. Control organoids were left untreated. After Wnt-surrogate addition, organoids were subjected to time-lapse imaging. To minimize technical artifacts, we adhered strictly to standardized procedures throughout organoid preparation and imaging. All seeding was performed at the same time of day, and both seeding and cell processing were carried out by the same investigator to avoid operator-dependent variability. Only reagents from reproducible sources were used, with careful attention to lot consistency and controlled freeze-thaw cycles (each aliquot was frozen once only). During imaging, temperature control was used to prevent fluctuations in culture conditions. In addition to these measures, organoids were subjected to manual quality control post-seeding to identify and exclude wells with contamination or lack of aggregation. No further adjustments (e.g., for illumination artifacts) were performed, as no systematic illumination issues were observed either within single images or over time.

## Organoid time-lapse imaging

Automated widefield time-lapse imaging of organoids was performed using the ACQUIFER Imaging Machine (ACQUIFER Imaging GmbH, Heidelberg, Germany) [33]. Organoids were placed in 96-well plates and incubated at 26 °C in the machine's plate holder, without $CO_2$ control. To prevent evaporation while enabling gas exchange, the plates were sealed with a Moisture Barrier Seal 96 (Azenta US; Cat#: 4ti-0516/96). Brightfield images of each organoid were captured every 30 min over a 72-hour period (144 time points) using 70% illumination, a 20 ms exposure time, and a 10× objective (NA 0.3, CFI Plan Fluor10X). Images were acquired as z-stacks with five slices, spaced 50 μm apart, around the automatically determined focal plane, resulting in a total of 69,120 brightfield images per experimental replicate for 96 organoids. Z-stacks were used for expert ground truth definition of tissue outcomes in the organoids as well as for generation of sum- and maximum-intensity z-projections, yet only the middle slice (slice 3/5) was used for image analysis and training of the models shown in the main figures. Following time-lapse imaging, organoids were imaged via stereomicroscopy (Nikon SMZ18 with a Digital Sight DS-Ri1 camera) to confirm RPE development.

Time-lapse recordings underwent manual quality checks, and wells exhibiting visible contamination were removed from further analysis and machine learning model training. Additionally, any time points where organoids were either out of frame or out of focus, as well as subsequent time points for the same organoid, were also excluded. As each well only contained one organoid, there were no additional steps necessary for organoid tracking or instance segmentation.

## Image ground truth annotation for machine learning model training

For training the machine learning models, time-lapse images of organoids were initially annotated for the presence of specific TOI in the organoids (RPE and lens) by two experts in medaka retinal organoid biology, working independently. Brightfield images, adjusted for contrast and brightness from the final time point in the automated widefield microscopy (time point 144th loop, 72 hours), were used for annotation for lenses, while stereomicroscopic images served as the gold standard for RPE detection and quantification (in concordance with the automated widefield microscopy). Annotations with consensus were assigned a high confidence score, while those with disagreement received a low confidence score. The decision on which expert annotation to include in the machine learning model training in case of a low confidence score was determined by a coin flip.

## Organoid classification by a human expert panel

Metadata from all experiments were merged and restricted to images corresponding to the middle slice (3/5). Loops were binned into six equal timeframes (24 loops covering 12h each). For every experiment-well pair and timeframe, one image was assigned to each of six annotators. When a group contained ≥6 unique images, images were sampled without replacement; smaller pools were sampled with replacement, so the same file could appear across annotators. Sanity checks confirmed that each annotator received exactly one image per available timeframe for every well, all annotators had the same load (5,271 images; 31,626 total) and no file repeated within an annotator. The histogram of loop indices in the final task list closely tracked the source distribution, indicating that the temporal profile was preserved. Unavoidable repeats resulted in 122 duplicate assignments overall, which were treated as individual data points. Final F1 scores (average = weighted) were calculated per experiment and timeframe and displayed as a mean in the respective figures.

## Tissue outcome of interest quantification and classification

To quantify RPE areas, RO were first segmented in stereomicroscopic images using Yen's global thresholding [34]. Regions of interest (ROIs) were defined based on the segmented organoid areas and applied to the 8-bit converted raw images. Minimum–Maximum re-scaling of all pixel values within the ROI was used to enhance the contrast between RPE and non-RPE pixel values. Re-scaled images were then subjected to Sauvola's local thresholding method, with a radius of 15 pixels, a $k$ value of 0.5, and the recommended r value of 128 [35]. The total area of all foreground pixels in the resulting binary images was then measured. For lens area quantification, lens areas were manually measured from slice 1, 3, or 5 of the acquired z-stack, depending on the focus plane of the lenses in the respective organoids, of the time-lapse brightfield images at the last time point (144th loop; 72 hours) using the oval tool in ImageJ [36]. All lens areas were measured as circles. Since lenses in medaka RO are already transparent by day 4, their areas were assumed to correspond to the central cross-section of the mostly spherical lenses. This allowed for reliable measurements when comparing between samples, even if the lenses were positioned differently within the organoids.

Quantitative image analysis was used to classify organoids into four groups for machine learning model training. Group 0 included organoids without development of the TOI, while groups 1, 2, and 3 represented low, medium, and high levels of tissue sizes, respectively. Cut-off values for these groups were determined by calculating the 33rd and 66th percentiles of phenotype measurements from the entire training dataset. For RPE amounts, group 1 (low) included organoids with RPE areas below 4,541.73 $\mu m^2$, group 2 (medium) had RPE areas between 4,541.73 $\mu m^2$ and 7,548.51 $\mu m^2$, and group 3 (high) consisted of organoids with RPE areas above 7,548.51 $\mu m^2$. A similar approach was applied to lens sizes, using cut-off values of 16,324.85 $\mu m^2$ and 29,083.23 $\mu m^2$. This classification method ensured that groups were objectively defined based on the actual distribution of quantitative tissue outcome data, resulting in equally sized groups for a balanced dataset for machine learning model training. To address the possibility that discretization into four outcome categories might obscure signals near bin boundaries, we evaluated prediction performance as a function of the distance between each sample and the center of its assigned bin. For the classical machine learning classifiers, we observed that

F1 scores tended to be higher for samples located near the bin edges compared to those near the bin centers. By contrast, the CNNs exhibited a more balanced performance across the full range of distances, with no systematic differences between center- and edge-proximal samples. These findings suggest that discretization did not impair model performance in either framework and that the CNNs, in particular, are robust to potential artifacts introduced by binning (S26 and S27 Figs).

## Organoid segmentation for image analysis and machine learning model training

In order to train a CNN to segment organoids, 841 organoid images were masked manually using Fiji [36] (v. 2.14.0/1.54f). The images and corresponding masks were read and downsampled to 512x512 px utilizing the INTER_AREA method of the open-cv package (v. 4.10.0). The dataset was subsequently split into a training set (90%) and a validation set (10%). A pre-trained version of DeepLabV3 with a ResNet101 backbone [37] was trained with an initial learning rate of $3 \times 10^{-4}$ and a batch size of 16 images for 500 epochs using the BCEWithLogitsLoss in the pytorch (v. 2.3.1) implementation. The classifier state with the lowest loss value on the validation set was saved and used for further segmentation tasks after manual confirmation of the accuracy on images not belonging to either the training or validation set.

## Large-scale time-lapse Image analysis

To measure morphological property metrics (termed morphometrics) of the organoids, the original images were downsampled to $512 \times 512$ px using the INTER_AREA method of the open-cv package and the mask was created using the trained segmentation-model. The mask was subsequently upsampled using the INTER_LINEAR implementation of open-cv and converted to a binary image with a threshold of 0.5 of the maximum pixel value. We extracted a broad set of quantitative image features from each organoid original image and mask using the regionprops_table function of *scikit-image* (v0.24.0). The feature space included geometric and intensity-based descriptors such as area, convex area, filled area, bounding box size, centroid coordinates, eccentricity, equivalent diameter, Euler number, extent, Feret's diameter, axis lengths, orientation, perimeter, solidity, and intensity statistics, as well as raw and weighted image moments. In addition to these standard region properties, we implemented several higher-level shape descriptors, including aspect ratio, roundness, compactness, circularity, form factor, effective diameter, and convexity. Custom features were implemented as follows: blur (variance of Laplace filter response within the mask), region-of-interest contrast (difference between maximum and minimum intensity inside the mask), image contrast (difference between maximum and minimum intensity of the full image), median intensity (median pixel intensity within the mask), modal value (most frequent pixel intensity within the mask), integrated density (mean intensity within the mask multiplied by area), raw integrated density (sum of pixel intensities within the mask), skewness (skew of the intensity distribution inside the mask), and kurtosis (kurtosis of the intensity distribution inside the mask). All features were computed on segmented organoid ROIs, and together capture organoid size, shape, and intensity distributions, forming the morphometrics feature space used for downstream machine learning models. Images where more than one mask were predicted or images where the mask would exceed a total diameter of $360 \times 360$ px were excluded. For full implementation details refer to the source code.

## Distance calculation and dimensionality reduction

Morphometrics were calculated as described above and Z-scaled. Subsequently, 20 principal components were calculated (PC space, Figs 2 and S2). For inter-organoid distances, pairwise Euclidean distances between all organoids at a given time point were calculated using the pdist function of scipy (v. 1.14.0, [38]). For intra-organoid distances, Euclidean distances between consecutive time points ($n$ versus $n+1$) were calculated for each organoid using the cdist function of *scipy*. Intra-organoid distances were quantile-capped at the 2.5th and 97.5th percentiles before plotting. tSNE dimensionality reduction was also performed using the *scikit-learn* (v1.5.1) implementation with default settings, but all distances shown in Fig 2C and 2D were computed exclusively in PC space. For the calculation of distances involving only organoids

without Wnt-surrogate treatment, the scaling was performed before data subsetting to preserve the numerical space. Data structure preservation by dimensionality reduction was performed by calculating the jaccard indices of 30 nearest neighbors in reduced space versus 30 nearest neighbors in PC space over time, as well as quantifying the section of the 30 nearest neighbors in both spaces that were derived from the same organoid (S3 Fig).

## Organoid classification using tabular metrics

Morphometrics were calculated as described above, Z-scaled and subsequently scaled to range from 0 to 1 to avoid negative values. In a first benchmark, classifiers in their sklearn (v. 1.5.1) implementation with standard settings were trained using leave-one-out training by reserving the data of one experiment as a test set. The classifiers were trained on 90% of organoids while 10% of organoids were reserved as an internal validation set. The validation and test data were scaled to the same range as the training data using the fitted scalers obtained from fitting on the training data only. This benchmark was performed for the binary classification of RPE and lenses as well as for the quantification of the tissue sizes thereof (S4–S6 and S15–S17 Figs). Selected classifiers for each readout were subjected to hyperparameter tuning based on their performance in the initial benchmark and known susceptibility to hyperparameter tuning. Hyperparameter tuning was performed using a custom implementation of the RandomHalvingSearch of sklearn with a reduction of factor of 3, a minimum_resource parameter of 1,000 and 5-fold cross-validation on the full dataset as described above (S4–S6 and S15–S17 Figs). The best performing parameters were saved and were used for the final evaluation. For full implementation details refer to the source code.

## Organoid classification using convolutional neural networks

Images were segmented as described above. Subsequently, a bounding box of 360 × 360 px was cropped using the mask and the resulting image was downsampled to 224 × 224 px. The images were scaled between 0 and 1 and stored together with the accompanying annotations in a custom class that would allow easy data access, splitting, and downsampling (for full implementation details refer to the source code). Training data were split in a ratio of 90:10, where the images of individual organoids were combined into groups such that all images from an individual organoid are assigned to either the training or the validation set, yielding a training- and validation-set, respectively. The test data derived from an independent experiment were read separately. For the training of CNNs, the pytorch (v2.3.1) implementations were used. Data were augmented using the albumentations (v1.4.12) package including rotations, distortions, random crops, rescalings and custom intensity adjustment, among others. We note that this augmentation step is indispensable, as omitting resulted in a severe overfit. The final normalization was performed only on the organoid pixels using a custom algorithm and used the same normalization values that were used for the initial model pretraining. Three models (DenseNet121 [39] ResNet50 [40] and MobileNetV3 [41]) were used in a pretrained state. Learning rates were determined beforehand for each readout using the pytorch-lr-finder package (https://github.com/davidtvs/pytorch-lr-finder). As a loss function and optimizer, CrossEntropyLoss and Adam were used in their pytorch implementations, respectively. Class weights for the loss function were calculated based on the distribution of the labels in the training set. To account for overfit, a weight-decay of 1e−4 was used for non-batchnorm layers for all models. Gradients were clipped to a maximum of 1.0. A learning rate scheduler was implemented that would decrease the learning rate by 50% with a patience of 7 epochs based on the F1-score on the test data. Models were trained for at least 30 epochs, and the best-performing model, judged by the loss on the validation set, was saved. For each epoch, the loss values on the respective data subsets as well as the F1-score were recorded for plotting. Baseline models were trained on shuffled labels. For evaluation, neural networks were instantiated and calibrated using temperature scaling [42], using a slightly modified implementation of one of the original authors (https://github.com/gpleiss/temperature_scaling). The models were used for evaluation and their predictions were weighted based on the best F1-score. The class with the highest output probability was then used as the predicted label and used for subsequent statistical analysis. For full implementation details refer to the source code.

## Generation of morphometrics-derived clusters

Images from the final time point were subjected to the morphometrics pipeline as described above. PCA with 15 principal components was performed on the scaled data. K-means clustering was used to calculate 4 distinct clusters that were used as classes to predict for the CNNs and machine learning classifiers. Datasets were annotated posthoc.

## Relevance backpropagation

To identify image regions driving network predictions, we computed saliency maps for each well and time point using three CNN architectures (DenseNet121, ResNet50, MobileNetV3_Large). For each model, a corresponding baseline network was included for comparison. We analyzed eight attribution methods covering different families: gradient-based (integrated gradients with noise tunnel (IG_NT, [43]), saliency with noise tunnel (SAL_NT, [44]), DeepLiftSHAP (DLS, [45])), as well as gradient-class activation (GradCAM (GC, [46]), Guided GradCAM (GGC, [46])) and occlusion based methods (Occlusion (OCC, [47]), Feature Ablation (FAB, [47]), Kernelshap (KSH, [45])). All algorithms were used as in the captum library (v.0.7.0, [48]). Input images were paired with a mean baseline reference masked to the organoid, and superpixel masks were generated with SLIC segmentation (50 segments, compactness = 0.1). Model-specific convolutional layers were used as the backpropagation targets. For every method and model, saliencies were computed both for the trained and baseline networks, reduced across channels (sum or mean), and stored in HDF5 files together with the corresponding image and mask. These files contained, for each well and loop, the attribution maps per model and per method in trained and baseline conditions. Each map was z-scored inside the organoid mask, and SLIC superpixels were computed within the mask for region analysis. We then derived four metrics. (1) Pairwise method agreement: within each model, we compared every method pair by converting maps into binary masks at the top 1%, 5%, and 10% of saliency inside the mask and taking the mean Dice across these thresholds. (2) Cross-model consistency: for each method, maps were converted to ranks over |saliency| inside the mask and compared between models using a fast Spearman-like correlation. (3) Region votes: for each model and loop, SLIC-derived superpixels were scored by their mean saliency per method, and a vote was assigned when a region's score fell within the top 10% for that image. (4) Entropy and drift over time: for each (model, method), probability maps of the top 10% of pixels were formed from saliency inside the mask to compute Shannon entropy (lower = more focused) and the frame-to-frame displacement of the center of mass (drift). For visualization, the DeepLiftShap algorithm was used using the segmented image and zero-pixels of the same shape as baselines, 0.001 stdevs and n_samples of 200. The trained DenseNet121 model for the respective readout was loaded and the attributions were calculated for each single image, of which the first layer was used for the final analysis. The absolute attribution values were normalized using captum and displayed in the figure. For full implementation details refer to the source code.

## Data visualization and statistical analyses

Plots were generated using the matplotlib (v. 3.9.1) and the seaborn (v. 0.13.2) libraries. For ANOVA calculation (S1 Fig), scipy was used. Sketches in Figs 1–3, and 5 were drawn using Inkscape 1.2.2 and Affinity Designer 1.10.5. F1-scores were calculated using the sklearn implementation with weighted averaging. To compare predictive performance across methods, we summarized F1-score curves over time as the area under the curve (AUC) for each experiment and classifier. The AUC was computed per experiment by trapezoidal integration of the F1-score trajectories over all available timepoints, normalized by the time span to yield the mean F1 across the time window. This resulted in one AUC value per experiment and classifier. Pairwise comparisons of classifiers were then performed using paired non-parametric Wilcoxon signed-rank tests (implemented in SciPy v1.14.0), which assessed whether the distribution of per-experiment AUC values differed significantly between methods. To account for multiple comparisons across classifier pairs, Holm–Bonferroni correction was applied to adjust $p$-values. The $p$-values have been provided as S1 File.

## Supporting information

**S1 Table. Number of images annotated per expert from the balanced subset for tissue visibility and prediction of outcomes.**
(XLSX)

**S2 Table. Dataset metrics across experiments. sd: standard deviation.**
(XLSX)

**S1 Note. Supplement results on the minimal organoid number for prediction tasks and comparison of the attribution methods to explain deep learning model decision-making.**
(DOCX)

**S1 Fig. Tissue-specific heterogeneity and tissue visibility in retinal organoids across development with increasing Wnt-surrogate concentrations. A** Development of retinal pigmented epithelium (RPE) and its amount by Wnt-surrogate concentration. Organoids were treated with the indicated concentrations of Wnt-surrogate and subjected to stereomicroscopy in order to detect the presence of RPE (left graph) and to measure its area (right graph). RPE development was observable after Wnt-surrogate treatment but absent in non-treated organoids over all conducted experiments. Notably, there was no consistent correlation of the concentration of the inducing agent and the rate (left graph) or amount (right graph) of RPE induction. Raw data of the figure plots have been deposited as Extended Data 9. **B** Development of lenses and their sizes by Wnt-surrogate concentration. Organoids were treated with the indicated concentrations of Wnt-surrogate. Lenses were detected from the time-lapse widefield microscopy images (left graph) and the area was measured (right graph). Lens development and lens sizes were found to be independent of the Wnt-surrogate-treatment and -concentration. The indicated *p*-values were derived from a one-way ANOVA over all four groups. Data points are color coded for the respective experiments. Raw data of the figure plots have been deposited as Extended Data 9. **C** RPE visibility over time and increased sensitivity of RPE detection by stereomicroscopy. Expert evaluation of the presence of RPE in each organoid over time (see Methods). F1-scores were calculated using the stereomicroscopy annotation as ground truth. RPE begins to visibly emerge around 68 h of organoid development. There is a notable difference between the stereomicroscopy derived ground truth and the human evaluation of the time-lapse images at later time points, suggesting an increased sensitivity of stereomicroscopy over the time-lapse images for the detection of visible RPE. The confusion matrix (right panel), summed over the last 6 hours of the imaging window, confirms the inferior sensitivity of human annotation from time-lapse images compared to stereomicroscopy. Raw data of the figure plots have been deposited as Extended Data 10. **D** Visibility of lenses in retinal organoids over time. The dataset was assembled as described in C and annotated by two experts for lens visibility in the respective images. The ground truth was defined by the same two independent researchers and F1-scores were calculated accordingly. Lenses begin to visibly emerge around 59 h of organoid development. Raw data of the figure plots have been deposited as Extended Data 10. **E/F** Distribution of RPE (E) and lens (F) classes across experiments. RPE and lens areas were quantified for each RPE and lens positive organoid in the dataset. There was a high variability in class distributions found across experiments, although culture conditions were kept uniform. For the class attributions refer to the Methods and Fig 1C. Raw data of the figure plots have been deposited as Extended Data 9.
(TIF)

**S2 Fig. Global morphological heterogeneity of retinal organoids across development between experiments. A** Retinal organoid images were analyzed using the image analysis platform (see Fig 2 and Methods). Data of the indicated experiments (E0XX) were subjected to t-SNE dimensionality reduction of the first 20 principal components obtained from the morphometrics data and colored by organoid-identity (respective left graph) and the time of acquisition (respective right graph). While the data points of organoid images from earlier time points cluster more closely, there is a substantial

divergence of organoids at later time points, suggesting increasing inter-individual differences of the morphologic characteristics over time. Raw data of the figure plots have been deposited as Extended Data 11. **B** Global morphological inter-organoid heterogeneity across experiments and time. The data correspond to the data shown in Fig 2D. Here, only organoids without Wnt-surrogate treatment were analyzed. Euclidean distances were calculated based on the first 20 principal components obtained from scaled morphometrics data for each organoid and plotted as the mean pairwise distance between all organoids for each time point. Notably, inter-organoid distance increased over time in an experiment-specific manner, indicating an increasing morphological divergence of the individual organoids over time. Raw data of the figure plots have been deposited as Extended Data 12. **C** Intra-organoid morphological changes over time. The data correspond to the data shown in Fig 2C. Here, only organoids without Wnt-surrogate treatment were analyzed. Euclidean distances were calculated based on 20 principal components of the morphometrics data for each organoid between time point $n$ and time point $n + 1$. The resulting metric reflects the amount of morphological changes during a time span of 30 min. While the relative changes are comparatively small at the beginning, the increase over time is suggesting more drastic morphological changes at later time points, consistent with the findings described in B and C. Raw data of the figure plots have been deposited as Extended Data 12.
(TIF)

**S3 Fig. Neighbor analysis confirms dimensionality reduction data structure.** The data directly correspond to Figs 2 and S2. **A–D** For each data point, 30 nearest neighbors were calculated in PC space (20 principal components derived from the morphometrics data, A/C) or in raw data space (scaled morphometrics data, B/D). Subsequently, 30 nearest neighbors were identified from dimensionality reductions in 2 dimensions (either TSNE or UMAP). The size of the intersection over the size of the union (jaccard index) was calculated over time. At earlier timepoints, jaccard scores are comparably low, confirming the dense cloud of data points and morphological similarity of the organoids. At later timepoints, the jaccard score increases, indicating a good data structure preservation by dimensionality reduction compared to raw distance measurements. Raw data of the figure plots have been deposited as Extended Data 13. **E/F** For each data point, 30 nearest neighbors have been calculated in either raw data space or PC space as described above. At each time point, the fraction of nearest neighbors corresponding to the same organoid has been quantified. At earlier timepoints, the structural similarity of individual organoids is high as indicated by the low overlap of nearest neighbors derived from the same individual organoid. At later timepoints, the fraction of nearest neighbors derived from the same organoid is almost 1, confirming the visual representation from Figs 2 and S2 and conclusion that organoids at later timepoints are morphologically diverse and more similar within itself than towards other organoids at the same timepoints. Raw data of the figure plots have been deposited as Extended Data 14.
(TIF)

**S4 Fig. Machine learning classifier benchmark and hyperparameter tuning for tissue emergence predictions on single slice images.** This analysis was performed on morphometrics data from single image slices. **A** The indicated classifiers were trained by cross-validation, using the indicated experiment as a test set, and scored using the F1 metric (y-axis) for the prediction of the presence and absence of RPE. Selected classifiers were subjected to hyperparameter tuning first (tuned). Raw data of the figure plots have been deposited as Extended Data 15. **B** The indicated classifiers were trained and evaluated as in A, but for the emergence of lenses. Raw data of the figure plots have been deposited as Extended Data 16.
(TIF)

**S5 Fig. Machine learning classifier benchmark and hyperparameter tuning for tissue emergence predictions on sum-intensity z-projection images.** This analysis was performed on morphometrics data from sum-intensity z-projections of all acquired 5 image slices. **A** The indicated classifiers were trained by cross-validation, using the indicated experiment as a test set, and scored using the F1 metric (y-axis) for the prediction of the presence and absence of

RPE. Selected classifiers were subjected to hyperparameter tuning first (tuned). Raw data of the figure plots have been deposited as Extended Data 17. **B** The indicated classifiers were trained and evaluated as in A, but for the emergence of lenses. Raw data of the figure plots have been deposited as Extended Data 18.
(TIF)

**S6 Fig. Machine learning classifier benchmark and hyperparameter tuning for tissue emergence predictions on maximum-intensity z-projection images.** This analysis was performed on morphometrics data from maximum-intensity z-projections of all 5 image slices. **A** The indicated classifiers were trained by cross-validation, using the indicated experiment as a test set, and scored using the F1 metric (y-axis) for the prediction of the presence and absence of RPE. Selected classifiers were subjected to hyperparameter tuning first (tuned). Raw data of the figure plots have been deposited as Extended Data 19. **B** The indicated classifiers were trained and evaluated as in A, but for the emergence of lenses. Raw data of the figure plots have been deposited as Extended Data 20.
(TIF)

**S7 Fig. Prediction of tissue emergence by tabular image analysis data obtained from single slice images.** Machine learning classifiers were evaluated on the ability to predict RPE emergence **(A, B)** and lens emergence **(C, D)** on the validation (left graph) and test (right graph) data sets (for the data partitioning strategy refer to Fig 3A and Methods). **A/C**: The data correspond directly to the data shown in Fig 3B (RPE emergence) and Fig 3C (lens emergence) but are split for the individual experiments. Raw data of the figure plots have been deposited as Extended Data 21 and 23, respectively. **B/D**: Confusion matrices. The data correspond to A and C, respectively. The x-axis denotes the respective imaging time points while the y-axes show the relative percentage of true-positive, true-negative, false-positive and false-negative predictions as indicated. Raw data of the figure plots have been deposited as Extended Data 22 and 24, respectively. Predictions were calculated using the function 'get_classification_f1_data' of the module orgAInoid.figures.figure_data_generation (compare source code).
(TIF)

**S8 Fig. Deep-learning aided prediction of RPE and lens tissue emergence in retinal organoids from sum- and maximum-intensity image z-projections.** Prediction of RPE emergence (**A/C**) and lens emergence (**B/D**) in images generated from sum- (**A/B**) and maximum-intensity z-projections (**C/D**). Sum- and maximum-intensity z-projection of the images did not gain significant performance enhancements compared to single-slice analysis (compare to Fig 3). QDA: Quadratic Discriminant Analysis; HGBC: Histogram Gradient Boosting Classifier. Raw data of the figure plots have been deposited as Extended Data 25–28 (A–D). Predictions were calculated using the function 'get_classification_f1_data' of the module orgAInoid.figures.figure_data_generation (compare source code).
(TIF)

**S9 Fig. Prediction of tissue emergence by tabular image analysis data obtained from sum-intensity z-projected images.** Machine learning classifiers were evaluated on the ability to predict RPE emergence (A, B) and lens emergence (C, D) on the validation (left graph) and test (right graph) data sets (for the data partitioning strategy refer to Fig 3A and Methods). **A/C**: The data correspond directly to the data shown in S8 Fig but are split for the individual experiments. Raw data of the figure plots have been deposited as Extended Data 29 and 31, respectively. **B/D**: Confusion matrices. The data correspond to A and C, respectively. The x-axis denotes the respective imaging time points while the y-axes show the relative percentage of true-positive, true-negative, false-positive and false-negative predictions as indicated. Raw data of the figure plots have been deposited as Extended Data 30 and 32, respectively. Predictions were calculated using the function 'get_classification_f1_data' of the module orgAInoid.figures.figure_data_generation (compare source code).
(TIF)

**S10 Fig. Prediction of tissue emergence by tabular image analysis data obtained from maximum-intensity z-projected images.** Machine learning classifiers were evaluated on the ability to predict RPE emergence (A, B) and lens emergence (C, D) on the validation (left graph) and test (right graph) data sets (for the data partitioning strategy refer to Fig 3A and Methods). **A/C**: The data correspond directly to the data shown in S8 Fig but are split for the individual experiments. Raw data of the figure plots have been deposited as Extended Data 33 and 35, respectively. **B/D**: Confusion matrices. The data correspond to A and C, respectively. The x-axis denotes the respective imaging time points while the y-axes show the relative percentage of true-positive, true-negative, false-positive and false-negative predictions as indicated. Raw data of the figure plots have been deposited as Extended Data 34 and 36, respectively. Predictions were calculated using the function 'get_classification_f1_data' of the module orgAInoid.figures.figure_data_generation (compare source code).
(TIF)

**S11 Fig. Training and evaluation of CNNs for RPE and lens emergence predictions.** CNNs were trained to predict the emergence of RPE (**A–C**) and lenses (**D–F**) from time-lapse single-slice images for the indicated amount of epochs and scored for the F1-metric in the training set (left graph), the validation set (middle graph) and the test set (right graph). The architecture of the respective CNN is noted within the respective title. Raw data of the figure plots have been deposited as Extended Data 37.
(TIF)

**S12 Fig. Prediction of tissue emergence by deep learning.** Deep learning classifiers were evaluated on the ability to predict RPE emergence (A, B) and lens emergence (C, D) on the validation (left graph) and test (right graph) data sets (for the data partitioning strategy refer to Fig 3A and Methods). **A/C**: The data correspond directly to the data shown in Fig 3B (RPE emergence) and Fig 3C (lens emergence) but are split for the individual experiments. Raw data of the figure plots have been deposited as Extended Data 38 and 40, respectively. **B/D**: Confusion matrices. The data correspond to A and C, respectively. The x-axis denotes the respective imaging time points while the y-axes show the relative percentage of true-positive, true-negative, false-positive and false-negative predictions as indicated. Raw data of the figure plots have been deposited as Extended Data 39 and 41, respectively. Predictions were calculated using the function 'get_classification_f1_data' of the module orgAInoid.figures.figure_data_generation (compare source code).
(TIF)

**S13 Fig. Prediction of tissue emergence by deep learning on images derived from sum-intensity z-projection.** Deep learning classifiers were evaluated on the ability to predict RPE emergence (A, B) and lens emergence (C, D) on the validation (left graph) and test (right graph) data sets (for the data partitioning strategy refer to Fig 3A and Methods). **A/C**: The data correspond directly to the data shown in S8 Fig, but are split for the individual experiments. Raw data of the figure plots have been deposited as Extended Data 42 and 44, respectively. **B/D**: Confusion matrices. The data correspond to A and C, respectively. The x-axis denotes the respective imaging time points while the y-axes show the relative percentage of true-positive, true-negative, false-positive and false-negative predictions as indicated. Raw data of the figure plots have been deposited as Extended Data 43 and 45, respectively. Predictions were calculated using the function 'get_classification_f1_data' of the module orgAInoid.figures.figure_data_generation (compare source code).
(TIF)

**S14 Fig. Prediction of tissue emergence by deep learning on images derived from maximum projection.** Deep learning classifiers were evaluated on the ability to predict RPE emergence (A, B) and lens emergence (C, D) on the validation (left graph) and test (right graph) data sets (for the data partitioning strategy refer to Fig 3A and Methods). **A/C**: The data correspond directly to the data shown in S8 Fig but are split for the individual experiments. Raw data of the figure

plots have been deposited as Extended Data 46 and 48, respectively. **B/D**: Confusion matrices. The data correspond to A and C, respectively. The x-axis denotes the respective imaging time points while the y-axes show the relative percentage of true-positive, true-negative, false-positive and false-negative predictions as indicated. Raw data of the figure plots have been deposited as Extended Data 47 and 49, respectively. Predictions were calculated using the function 'get_classification_f1_data' of the module orgAInoid.figures.figure_data_generation (compare source code). (TIF)

**S15 Fig. Machine learning classifier benchmark and hyperparameter tuning for tissue size predictions. A** The indicated classifiers were trained by cross-validation, using the indicated experiment as a test set, and scored using the F1 metric (y-axis) for the prediction of RPE area class. Selected classifiers were subjected to hyperparameter tuning first (tuned). Raw data of the figure plots have been deposited as Extended Data 50. **B** The indicated classifiers were trained and evaluated as in A, but for the class of lens area. Raw data of the figure plots have been deposited as Extended Data 51. (TIF)

**S16 Fig. Machine learning classifier benchmark and hyperparameter tuning for tissue size predictions on sum-intensity z-projection images. A** The indicated classifiers were trained by cross-validation, using the indicated experiment as a test set, and scored using the F1 metric (y-axis) for the prediction of RPE area class. Selected classifiers were subjected to hyperparameter tuning first (tuned). Raw data of the figure plots have been deposited as Extended Data 52. **B** The indicated classifiers were trained and evaluated as in A, but for the class of lens area. Raw data of the figure plots have been deposited as Extended Data 53. (TIF)

**S17 Fig. Machine learning classifier benchmark and hyperparameter tuning for tissue size predictions on maximum-intensity z-projection images. A** The indicated classifiers were trained by cross-validation, using the indicated experiment as a test set, and scored using the F1 metric (y-axis) for the prediction of RPE area class. Selected classifiers were subjected to hyperparameter tuning first (tuned). Raw data of the figure plots have been deposited as Extended Data 54. **B** The indicated classifiers were trained and evaluated as in A, but for the class of lens area. Raw data of the figure plots have been deposited as Extended Data 55. (TIF)

**S18 Fig. Prediction of tissue sizes by tabular image analysis data.** Machine learning classifiers were evaluated on the ability to predict RPE areas (A, B) and lens areas (C, D) on the validation (left graph) and test (right graph) data sets (for the data partitioning strategy refer to Fig 3A and Methods). **A/C**: The data correspond directly to the data shown in Fig 4A (RPE emergence) and Fig 4B (lens emergence) but are split for the individual experiments. Raw data of the figure plots have been deposited as Extended Data 56 and 58, respectively. **B/D**: Confusion matrices. The data correspond to A and C, respectively. The x-axis denotes the respective imaging time points while the y-axes show the relative percentage of true-positive, true-negative, false-positive and false-negative predictions as indicated. Raw data of the figure plots have been deposited as Extended Data 57 and 59, respectively. Predictions were calculated using the function 'get_classification_f1_data' of the module orgAInoid.figures.figure_data_generation (compare source code). (TIF)

**S19 Fig. Deep-learning aided prediction of RPE and lens tissue sizes in retinal organoids.** Prediction of RPE (**A/C**) and lens (**B/D**) tissue sizes by deep learning well before visibility on images derived from sum- (**A/B**) or maximum-intensity z-projections (**C/D**). Sum- and maximum-intensity z-projection of the images did not gain significant performance enhancements compared to single-slice analysis (compare Fig 4). QDA: Quadratic Discriminant Analysis; HGBC: Histogram Gradient Boosting Classifier. Raw data of the figure plots have been deposited as Extended Data 60–63(A–D).

Predictions were calculated using the function 'get_classification_f1_data' of the module orgAInoid.figures.figure_data_generation (compare source code).
(TIF)

**S20 Fig. Prediction of tissue sizes by tabular image analysis data obtained from sum-intensity z-projection images.** Machine learning classifiers were evaluated on the ability to predict RPE areas (A, B) and lens areas (C, D) on the validation (left graph) and test (right graph) data sets (for the data partitioning strategy refer to Fig 3A and Methods). **A/C**: The data correspond directly to the data shown in S19 Fig but are split for the individual experiments. Raw data of the figure plots have been deposited as Extended Data 64 and 66, respectively. **B/D**: Confusion matrices. The data correspond to A and C, respectively. The x-axis denotes the respective imaging time points while the y-axes show the relative percentage of true-positive, true-negative, false-positive and false-negative predictions as indicated. Raw data of the figure plots have been deposited as Extended Data 65 and 67, respectively. Predictions were calculated using the function 'get_classification_f1_data' of the module orgAInoid.figures.figure_data_generation (compare source code).
(TIF)

**S21 Fig. Prediction of tissue sizes by tabular image analysis data obtained from maximum-intensity z-projection images.** Machine learning classifiers were evaluated on the ability to predict RPE areas (A, B) and lens areas (C, D) on the validation (left graph) and test (right graph) data sets (for the data partitioning strategy refer to Fig 3A and Methods). **A/C**: The data correspond directly to the data shown in S19 Fig but are split for the individual experiments. Raw data of the figure plots have been deposited as Extended Data 68 and 70, respectively. **B/D**: Confusion matrices. The data correspond to A and C, respectively. The x-axis denotes the respective imaging time points while the y-axes show the relative percentage of true-positive, true-negative, false-positive and false-negative predictions as indicated. Raw data of the figure plots have been deposited as Extended Data 69 and 71, respectively. Predictions were calculated using the function 'get_classification_f1_data' of the module orgAInoid.figures.figure_data_generation (compare source code).
(TIF)

**S22 Fig. Training and evaluation of CNNs for RPE and lens area predictions.** CNNs were trained to predict the areas of RPE (**A–C**) and lenses (**D–F**) from time-lapse images for the indicated amount of epochs and scored for the F1 metric in the training set (left graph), the validation set (middle graph) and the test set (right graph). The architecture of the respective CNN is noted within the title. Raw data of the figure plots have been deposited as Extended Data 72.
(TIF)

**S23 Fig. Prediction of tissue size by deep learning.** Deep learning classifiers were evaluated on the ability to predict RPE areas (A, B) and lens areas (C, D) on the validation (left graph) and test (right graph) data sets (for the data partitioning strategy refer to Fig 3A and Methods). **A/C**: The data correspond directly to the data shown in Fig 4A (RPE emergence) and Fig 4B (lens emergence) but are split for the individual experiments. Raw data of the figure plots have been deposited as Extended Data 73 and 75, respectively. **B/D**: Confusion matrices. The data correspond to A and C, respectively. The x-axis denotes the respective imaging time points while the y-axes show the relative percentage of true-positive, true-negative, false-positive and false-negative predictions as indicated. Raw data of the figure plots have been deposited as Extended Data 74 and 76, respectively. Predictions were calculated using the function 'get_classification_f1_data' of the module orgAInoid.figures.figure_data_generation (compare source code).
(TIF)

**S24 Fig. Prediction of tissue size by deep learning on sum-intensity z-projected images.** Deep learning classifiers were evaluated on the ability to predict RPE areas (A, B) and lens areas (C, D) on the validation (left graph) and test (right graph) data sets (for the data partitioning strategy refer to Fig 3A and Methods). **A/C**: The data correspond directly to

the data shown in S19 Fig but are split for the individual experiments. Raw data of the figure plots have been deposited as Extended Data 77 and 79, respectively. **B/D**: Confusion matrices. The data correspond to A and C, respectively. The x-axis denotes the respective imaging time points while the y-axes show the relative percentage of true-positive, true-negative, false-positive and false-negative predictions as indicated. Raw data of the figure plots have been deposited as Extended Data 78 and 80, respectively. Predictions were calculated using the function 'get_classification_f1_data' of the module orgAInoid.figures.figure_data_generation (compare source code).
(TIF)

**S25 Fig. Prediction of tissue size by deep learning on maximum-intensity z-projected images.** Deep learning classifiers were evaluated on the ability to predict RPE areas (A, B) and lens areas (C, D) on the validation (left graph) and test (right graph) data sets (for the data partitioning strategy refer to Fig 3A and Methods). **A/C**: The data correspond directly to the data shown in S19 Fig but are split for the individual experiments. Raw data of the figure plots have been deposited as Extended Data 81 and 83, respectively. **B/D**: Confusion matrices. The data correspond to A and C, respectively. The x-axis denotes the respective imaging time points while the y-axes show the relative percentage of true-positive, true-negative, false-positive and false-negative predictions as indicated. Raw data of the figure plots have been deposited as Extended Data 82 and 84, respectively. Predictions were calculated using the function 'get_classification_f1_data' of the module orgAInoid.figures.figure_data_generation (compare source code).
(TIF)

**S26 Fig. F1 score distribution over bin center distance for convolutional neural networks. A–D** Weighted F1 scores for RPE area (**A/B**) and lens size (**C/D**) in validation (**A/C**) and test (**B/D**) organoids. CNNs demonstrate a more balanced performance across bin distances, without systematic differences between center- and edge-proximal samples. Raw data of the figure plots have been deposited as Extended Data 85.
(TIF)

**S27 Fig. F1 score distribution over bin center distance for classical machine learning classifiers. A–D** Weighted F1 scores for RPE area (**A/B**) and lens size (**C/D**) in validation (**A/C**) and test (**B/D**) organoids. The x-axis indicates the normalized distance from the bin center (0 = center, 1 = edge), and the y-axis shows the weighted F1 score. Each line represents one test experiment (E001–E011). Classifiers show a tendency for higher F1 scores near bin edges. Raw data of the figure plots have been deposited as Extended Data 86.
(TIF)

**S28 Fig. Number of experiments needed for an accurate prediction of TOI.** Number of experiments needed for an accurate prediction of RPE (**A**) and lens (**B**) emergence, and RPE (**C**) and lens (**D**) area. The experiments corresponding to Figs 3 and 4 were repeated with the indicated number of total experiments (x-axis) used for training. We chose MobileNetV3_Large as the CNN due to computational resources, which was trained for exactly one epoch (left graph), as we did not observe a striking increase in validation accuracy after a higher number of epochs (compare S11 and S22 Figs). The machine learning classifiers (right graph) were used similarly to Figs 3 and 4, dependent on the classification task. While the curves plateaued at approximately 6 experiments for the classification by neural networks, we could not observe a similar trend for the machine learning classifiers, indicating that tabular data guided classification needs fewer training experiments for a comparable accuracy on the test set. Raw data of the figure plots have been deposited as Extended Data 87 and 88(deep learning and morphometrics, respectively).
(TIF)

**S29 Fig. Pairwise agreement of attribution methods within convolutional neural networks.** Saliency maps were generated for three CNN architectures (DenseNet121, MobileNetV3_Large, ResNet50) using eight attribution

methods: integrated gradients (IG_NT), simple saliency (SAL_NT), DeepLIFT SHAP (DLS), Grad-CAM (GC), guided Grad-CAM (GGC), smooth occlusion (OCC), feature ablation (FAB), and kernel SHAP (KSH). For each method pair within a given model, Dice coefficients were calculated by thresholding the top 1%, 5%, and 10% of saliency values inside the organoid mask and averaging across thresholds. Shown are mean Dice coefficients over time for each readout (A, RPE emergence; B, lens emergence; C, RPE area; D, lens size). Most method pairs displayed low overlap close to baseline levels, while Grad-CAM and guided Grad-CAM consistently achieved higher agreement. Raw data of the figure plots have been deposited as Extended Data 89.
(TIF)

**S30 Fig. Representative saliency maps for RPE emergence.** Shown are saliency maps for a representative organoid (experiment E001, well B003) across three convolutional neural network architectures (DenseNet121, ResNet50, Mobile-NetV3_Large) and three time points (0 h, 15 h, 72 h). Columns display attribution results for eight methods: Integrated Gradients (IG_NT), Saliency (SAL_NT), DeepLiftSHAP (DLS), Grad-CAM (GC), Guided Grad-CAM (GGC), Smooth Occlusion (OCC), Feature Ablation (FAB), and Kernel SHAP (KSH). Input images with organoid masks (green) are shown in the first column of each block. Saliencies were z-scored within the organoid mask, and heatmaps are displayed on a common color scale (blue = low, red = high). The comparison highlights differences in how CNN architectures and attribution methods assign relevance, with CAM-based approaches showing more localized hotspots, perturbation-based methods distributing relevance across discrete patches, and gradient-based methods yielding broader and more diffuse patterns.
(TIF)

**S31 Fig. Representative saliency maps for Lens emergence.** Shown are saliency maps for a representative organoid (experiment E001, well B003) across three convolutional neural network architectures (DenseNet121, ResNet50, Mobile-NetV3_Large) and three time points (0 h, 15 h, 72 h). Columns display attribution results for eight methods: Integrated Gradients (IG_NT), Saliency (SAL_NT), DeepLiftSHAP (DLS), Grad-CAM (GC), Guided Grad-CAM (GGC), Smooth Occlusion (OCC), Feature Ablation (FAB), and Kernel SHAP (KSH). Input images with organoid masks (green) are shown in the first column of each block. Saliencies were z-scored within the organoid mask, and heatmaps are displayed on a common color scale (blue = low, red = high). The comparison highlights differences in how CNN architectures and attribution methods assign relevance, with CAM-based approaches showing more localized hotspots, perturbation-based methods distributing relevance across discrete patches, and gradient-based methods yielding broader and more diffuse patterns.
(TIF)

**S32 Fig. Representative saliency maps for RPE area.** Shown are saliency maps for a representative organoid (experiment E001, well B003) across three convolutional neural network architectures (DenseNet121, ResNet50, MobileNetV3_Large) and three time points (0 h, 15 h, 72 h). Columns display attribution results for eight methods: Integrated Gradients (IG_NT), Saliency (SAL_NT), DeepLiftSHAP (DLS), Grad-CAM (GC), Guided Grad-CAM (GGC), Smooth Occlusion (OCC), Feature Ablation (FAB), and Kernel SHAP (KSH). Input images with organoid masks (green) are shown in the first column of each block. Saliencies were z-scored within the organoid mask, and heatmaps are displayed on a common color scale (blue = low, red = high). The comparison highlights differences in how CNN architectures and attribution methods assign relevance, with CAM-based approaches showing more localized hotspots, perturbation-based methods distributing relevance across discrete patches, and gradient-based methods yielding broader and more diffuse patterns.
(TIF)

**S33 Fig. Representative saliency maps for Lens sizes.** Shown are saliency maps for a representative organoid (experiment E001, well B003) across three convolutional neural network architectures (DenseNet121, ResNet50, MobileNetV3_Large) and three time points (0 h, 15 h, 72 h). Columns display attribution results for eight methods: Integrated Gradients (IG_NT), Saliency (SAL_NT), DeepLiftSHAP (DLS), Grad-CAM (GC), Guided Grad-CAM (GGC), Smooth Occlusion

(OCC), Feature Ablation (FAB), and Kernel SHAP (KSH). Input images with organoid masks (green) are shown in the first column of each block. Saliencies were z-scored within the organoid mask, and heatmaps are displayed on a common color scale (blue = low, red = high). The comparison highlights differences in how CNN architectures and attribution methods assign relevance, with CAM-based approaches showing more localized hotspots, perturbation-based methods distributing relevance across discrete patches, and gradient-based methods yielding broader and more diffuse patterns. (TIF)

**S34 Fig. Cross-model consistency of attribution methods.** Saliency maps were generated for three CNN architectures (DenseNet121, MobileNetV3_Large, ResNet50) using eight attribution methods: integrated gradients (IG_NT), simple saliency (SAL_NT), DeepLIFT SHAP (DLS), Grad-CAM (GC), guided Grad-CAM (GGC), smooth occlusion (OCC), feature ablation (FAB), and kernel SHAP (KSH). For each method, saliency maps were converted to ranked pixel values (absolute saliency inside the organoid mask) and compared across models using a Spearman-like correlation. Shown are mean correlations ± SEM over time for each readout (A, RPE emergence; B, lens emergence; C, RPE area; D, lens size). Gradient-based approaches such as DeepLIFT SHAP and integrated gradients achieved the highest and most stable consistency across models, while CAM- and perturbation-based methods were less consistent. Raw data of the figure plots have been deposited as Extended Data 90. (TIF)

**S35 Fig. Entropy diffusion of attribution methods across CNN architectures.** Entropy values of saliency maps were computed for three convolutional neural network architectures (DenseNet121, MobileNetV3_Large, ResNet50) and four readouts: (A) RPE emergence, (B) lens emergence, (C) RPE area, and (D) lens size. Eight attribution methods were compared: DeepLIFT SHAP (DLS), feature ablation (FAB), Grad-CAM (GC), guided Grad-CAM (GGC), integrated gradients (IG_NT), kernel SHAP (KSH), smooth occlusion (OCC), and simple saliency (SAL_NT). Entropy was calculated on the top 10% of saliency values inside the organoid mask (lower entropy = more focused attribution). Grad-CAM and guided Grad-CAM consistently showed the lowest entropy and further decreased over time, indicating increasingly focused saliency. FAB and kernel SHAP also decreased in several settings, while gradient-based (DLS, IG_NT, SAL_NT) and occlusion-based (OCC) methods remained more diffuse with higher entropy. Raw data of the figure plots have been deposited as Extended Data 91. (TIF)

**S36 Fig. Regional voting analysis of attribution methods.** Saliency maps from three convolutional neural network architectures (DenseNet121, MobileNetV3_Large, ResNet50) were segmented into superpixels, and each method assigned votes to the top 10% most salient regions within the organoid mask. Shown is the fraction of regions per well that were consistently labeled by at least two (red), three (blue), or four (green) different attribution methods over time. Results are displayed for four readouts: (A) RPE emergence, (B) lens emergence, (C) RPE area, and (D) lens size. Agreement between methods was highest at early time points, with many regions receiving votes from multiple methods, but this overlap progressively declined as development advanced, indicating that saliency methods increasingly diverged in the regions they prioritized. Raw data of the figure plots have been deposited as Extended Data 92. (TIF)

**S37 Fig. Spatial drift of saliency map centers of mass.** The displacement of the saliency map center of mass between consecutive timepoints is shown for three convolutional neural network architectures (DenseNet121, MobileNetV3_Large, ResNet50) across four prediction tasks: (A) RPE emergence, (B) lens emergence, (C) RPE area, and (D) lens size. Each attribution method is represented separately (DLS: DeepLIFT SHAP, IG_NT: Integrated Gradients, SAL_NT: Simple Saliency, GC: Grad-CAM, GGC: Guided Grad-CAM, OCC: Smooth Occlusion, FAB: Feature Ablation, KSH: Kernel SHAP). Lower drift values indicate that relevance remains in similar image regions across time, while higher values reflect

more dynamic relocation of relevance. CAM-based methods (Grad-CAM, Guided Grad-CAM) and perturbation-based methods (Feature Ablation, Kernel SHAP) generally showed the strongest drift, whereas gradient-based methods (DeepLIFT SHAP, Integrated Gradients, Saliency) and Smooth Occlusion exhibited lower and more stable displacement over time. Raw data of the figure plots have been deposited as Extended Data 93.
(TIF)

**S1 File. *p*-value statistics of the comparison of the respective prediction curves.**
(CSV)

## Author contributions

**Conceptualization:** Cassian Afting, Joachim Wittbrodt, Tarik Exner.

**Funding acquisition:** Joachim Wittbrodt, Tarik Exner.

**Investigation:** Cassian Afting, Norin Bhatti, Christina Schlagheck, Encarnación Sánchez Salvador, Laura Herrera-Astorga, Rashi Agarwal, Risa Suzuki, Nicolaj Hackert, Hanns-Martin Lorenz, Lucie Zilova, Joachim Wittbrodt, Tarik Exner.

**Project administration:** Cassian Afting, Tarik Exner.

**Supervision:** Joachim Wittbrodt, Tarik Exner.

**Visualization:** Cassian Afting, Tarik Exner.

**Writing – original draft:** Cassian Afting, Tarik Exner.

**Writing – review & editing:** Cassian Afting, Norin Bhatti, Christina Schlagheck, Encarnación Sánchez Salvador, Laura Herrera-Astorga, Rashi Agarwal, Risa Suzuki, Nicolaj Hackert, Hanns-Martin Lorenz, Lucie Zilova, Joachim Wittbrodt, Tarik Exner.

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
