## [Editor Report · Decision Letter 0]

10 Nov 2025

Dear Dr Exner,

Thank you for submitting your revised manuscript via Review Commons entitled "Deep learning predicts tissue outcomes in retinal organoids" for consideration as a Research Article by PLOS Biology.

Your manuscript has now been evaluated by the PLOS Biology editorial staff as well as by an academic editor with relevant expertise and I am writing to let you know that we would like to send your revision back to the original reviewers.

However, before we can send your manuscript back to the reviewers, we need you to complete your submission by providing the metadata that is required for full assessment. To this end, please login to Editorial Manager where you will find the paper in the 'Submissions Needing Revisions' folder on your homepage. Please click 'Revise Submission' from the Action Links and complete all additional questions in the submission questionnaire.

Once your full submission is complete, your paper will undergo a series of checks in preparation for peer review. After your manuscript has passed the checks it will be sent out for review. To provide the metadata for your submission, please Login to Editorial Manager (https://www.editorialmanager.com/pbiology) within two working days, i.e. by Nov 12 2025 11:59PM.

Kind regards,

Ines

--

Ines Alvarez-Garcia, PhD

Senior Editor

PLOS Biology

---

## [Decision Letter · Decision Letter 1]

20 Dec 2025

Dear Dr Exner,

Thank you for your patience while we considered your revised manuscript entitled "Deep learning predicts tissue outcomes in retinal organoids" for publication as a Research Article at PLOS Biology. This revised version of your manuscript has been evaluated by the PLOS Biology editors, the Academic Editor and by two of the original reviewers from Review Commons.

Based on the reviews, we are likely to accept this manuscript for publication, provided you consider the recommendations suggested by Reviewer 3 for future work, which you could discuss in the text. We would also like to change the article type to Methods and Resources, as we do think it fits better that format, thus please select this article type from the dropdown menu when you submit the final revision. Please also make sure to address the data and other policy-related requests stated below my signature.

In addition, we would like you to consider a suggestion to improve the title:

"A deep learning-based computational pipeline predicts developmental outcome in retinal organoids”

We expect to receive your revised manuscript within two weeks.

*Published Peer Review History*

*Press*

Sincerely,

Ines

--

Ines Alvarez-Garcia, PhD

Senior Editor

PLOS Biology

DATA POLICY:

Fig. 1B, C; Fig. 2C, D; Fig. S1A-E; Fig. S2B, C; Fig. S3A-F; Fig. S4Am B; Fig. S5A, B; Fig. S6A, B; Fig. S11; Fig. S15A, B; Fig. S16A, B; Fig. S17A, B; Fig. S22; Fig. S26; Fig. S27; Fig. S28; Fig. S29; Fig. S34; Fig. S35; Fig. S36 and Fig. S37

For the figures that have predictions, please make sure that you add to the figure legend the link to the code used to generate the data. Apologies if I have missed some from the list above.

CODE POLICY

Reviewers' comments

Rev. 1: Constantin Pape - note that this reviewer has signed the review.

The revised manuscript addresses all my previous comments. I recommend it for publication in PLOS Biology

Rev. 3:

I would like to thank the authors for their thorough and transparent response to the previous round of review. The revised manuscript serves as a rigorous proof-of-concept for the application of deep learning in predicting organoid developmental trajectories. The authors have significantly strengthened the technical foundations of the work, addressed concerns regarding statistical rigor, and expanded the scope of their predictive framework.

While I am recommending acceptance based on the methodological soundness and the potential utility of the "Latent Determination Horizon" framework, I must note that the revision was unable to fully resolve two major limitations regarding generalizability and biological interpretability. These remaining issues do not preclude publication of this technical advance, but they do define the clear boundaries of the current study. Below, I outline the improvements that underpin my decision to accept, followed by specific recommendations for addressing the lingering limitations in future work.

1. Improvements and Rationale for Acceptance

The authors have successfully addressed several critical technical concerns raised in my initial report, significantly improving the robustness of the study:

Expansion of Predictive Scope (Figure 4C): The inclusion of the analysis predicting abstract morphological clusters at the final timepoint is a substantial improvement. This demonstrates that the model is not limited to recognizing specific tissue structures (like lens or RPE) but can capture global developmental trajectories early in the culture period. This generalization elevates the work from a simple binary classifier to a broader morphological forecasting tool.

Correction of Dimensionality Issues: I commend the authors for addressing the critique regarding Euclidean distances in high-dimensional space. The shift to calculating distances within a PCA-reduced space, validated by the Nearest Neighbor Jaccard analysis (Supplementary Figure S3), provides a much more mathematical sound basis for the claims regarding morphological divergence.

Statistical Rigor: The addition of formal statistical testing (paired Wilcoxon signed-rank tests with Holm-Bonferroni correction) provides the necessary quantitative evidence that the CNN ensemble significantly outperforms classical machine learning baselines.

Reproducibility: The public release of the full code repository and raw data on Zenodo/GitHub is a vital step. Given the niche nature of the experimental model, ensuring the community can inspect and adapt the code is essential for the method's translation to other systems.

2. Remaining Limitations and Recommendations for Future Work

While the revisions justify publication, the response indicates that Major Issues 1 (Generalizability) and 2 (Biological Interpretability) from my previous report remain largely unsolved due to inherent constraints of the study design.

A. Generalizability (The "Single-Lab" Constraint) The authors have acknowledged that they cannot validate the model on external datasets because the specific Oryzias latipes differentiation protocol is unique to their laboratory. Consequently, the model technically remains overfitted to a single laboratory’s specific experimental conditions, species, and imaging setup.

Future Recommendation: To transition this from a "single-lab demonstration" to a community tool, future work must prioritize domain adaptation. I strongly suggest establishing collaborations to generate parallel datasets using mammalian (mouse/human) organoids. Even small "few-shot" learning experiments—where a model pre-trained on Medaka data is fine-tuned with a small number of human organoid images—would be a powerful way to demonstrate true utility to the broader biomedical community.

B. Biological Interpretability (The "Black Box" Issue) The authors performed an exhaustive computational analysis using eight different attribution methods across three CNN architectures. However, the results were largely "negative" in terms of biological insight: the methods showed low consistency and failed to identify stable, human-interpretable features (e.g., specific texture patterns or precursor structures) driving the predictions.

Future Recommendation: It is clear that standard saliency maps (e.g., Grad-CAM, DeepLIFT) are insufficient for this type of diffused biological signal. For future studies, I recommend moving beyond pixel-attribution methods. Generative Approaches: Techniques involving Generative Adversarial Networks (GANs) or Variational Autoencoders (VAEs) could be used to "dream" or synthesize images that maximize the "RPE-positive" class score. Visualizing these exaggerated synthetic features might reveal the texture or shape cues the model is reacting to, which are likely too subtle or distributed for saliency maps to capture. Correlative Approaches: Correlating activation vectors from intermediate network layers with transcriptomic data (if available in the future) could help link the "black box" features to specific molecular programs, bridging the gap between pixel data and biological mechanisms. To conclude, this manuscript successfully establishes a robust computational framework for predicting organoid fate and defines a valuable "determination window" for future experimentation. Despite the "black box" nature of the predictions and the species-specific limitations, the methodological rigor of the revised analysis warrants acceptance.

---

## [Editor Report · Decision Letter 2]

5 Jan 2026

Dear Dr Exner,

Thank you for the submission of your revised Methods and Resources entitled "A deep learning-based computational pipeline predicts developmental outcome in retinal organoids" for publication in PLOS Biology. On behalf of my colleagues and the Academic Editor, Bon-Kyoung Koo, I am delighted to let you know that we can in principle accept your manuscript for publication, provided you address any remaining formatting and reporting issues. These will be detailed in an email you should receive within 2-3 business days from our colleagues in the journal operations team; no action is required from you until then. Please note that we will not be able to formally accept your manuscript and schedule it for publication until you have completed any requested changes.

PRESS

Sincerely,

Ines

--

Ines Alvarez-Garcia, PhD

Senior Editor

PLOS Biology
